



# 1    EARLINET lidar quality assurance tools

Volker Freudenthaler[1], Holger Linné[2], Anatoli Chaikovski[3], Dieter Rabus[1,a], Silke Groß[1,b]
[1]Meteorologisches Institut der Ludwig–Maximilians–Universität, Munich, Germany
[2]Max Planck Institute for Meteorology, Hamburg, Germany
[3]B.I. Stepanov Institute of Physics, National Academy of Sciences of Belarus, Minsk, Belarus
[a]retired
[b]now at: Deutsches Zentrum für Luft- und Raumfahrt, Institute of Atmospheric Physics, Oberpfaffenhofen,
Germany
*Correspondence to*: V. Freudenthaler (volker.freudenthaler@lmu.de)
**Abstract.** This paper describes the EARLINET quality assurance (QA) check-up tools for the hardware of
lidars, developed in the recent years to monitor and improve the quality of the lidar systems and of their
products. These check-up tools are the trigger-delay test, the Rayleigh-fit, the lidar test-pulse generator, the dark
measurement, the telecover test, and the polarisation calibration.

## 1    Introduction

The introductory paper of this special issue (Pappalardo et al. 2014) gives an overview of the development of the
European Aerosol Research Lidar Network, EARLINET, since its foundation in 2000, and a detailed
introduction to the present paper in section 3.1, wherefore it will not be repeated here. More than 20 lidar groups
work together since 2000 with a very heterogeneous field of lidars and calibration procedures. For lack of
standardised equipment and common quality assurance procedures - not only within EARLINET - there was a
need for standardisation in order to make the lidar products of the different systems comparable and to be able to
provide quality-assured data sets of network products for the characterisation of the European aerosol conditions
as function of height. Because an atmospheric standard target doesn't exist, apart from the far range Rayleigh
calibration described below, and because the main target of EARLINET was the tropospheric and boundary laser
aerosol in the near range of lidars, which was (and is) a critical range for old-style lidars developed in the age
and with the money of stratospheric ozone research, we started to develop standardised tests for the lidar's sub-
systems which should help to characterise and finally homogenise the performance of lidars also in the near
range. Although a direct lidar intercomparison with a reference lidar is one possibility, the reference lidar itself
has to be characterised first. Furthermore, such intercomparisons are expensive, need a mobile reference system,
or the lidar itself must be mobile, which always bears the risk of damage and misalignments during the transport,
and cannot be done frequently enough. Nevertheless, a direct lidar intercomparison with a reference system bears
a high credibility and often reveals to date unknown problems, wherefore it still is considered the ultimate test
after all others are passed. Several intercomparisons were conducted in the frame of EARLINET, over which
Wandinger et al. (2016) give an overview. In this paper we focus on self-testing check-up tools with the
emphasis that the tools should be cheap, simple, everywhere to use, and for a variety of lidar systems with
comparable results. Of course, the tools should address the main problems of near range and tropospheric aerosol
lidars. The uncertain trigger delay between the outgoing laser pulse and the start of the data recording, especially
with low resolution transient recorders, causes large uncertainties in the near range (section 2). The high dynamic





range of near to far range signals (see Figure 11) can cause a too high signal-to-noise ratio or signal distortions in
the far range (analogue signals) with uncertainty in the Rayleigh calibration and/or overload of the signal in the
near range (photon counting saturation). The Rayleigh-fit test to check the far range accuracy and also the laser
pointing alignment is described in section 3. For this test we use the Rayleigh signal, which is calculated with
radiosonde data of the air pressure and temperature. Because there are many different approximations in the
literature with various accuracy, we collect in App. 9.2 the equations and approximations we use and provide a
table of the scattering coefficients and linear depolarisation ratios for the most common lidar wavelengths for
comparison with the readers own calculations. The lidar pulse generator, with which the accuracy limits of
analogue signals can be tested, is described in section 4. Some laser trigger synchronous distortions of analogue
signals can be determined with the dark measurement (section 5) and subtracted from the normal signal. The
telecover test together with realistic raytracing simulations, described in section 6, enables to determine a lower
limit for the distance of full overlap between laser and telescope field of view, and to detect several other
misalignments of the receiver and transmitter optics. Finally, we treat the test of the relative calibration of two
depolarisation channels with the molecular depolarisation in section 7.

## 2    Trigger delay

A trigger-delay between the actual laser pulse emission and the assumed zero range of the signal recording (zero-
bin) can cause large errors in the near range signal up to about 1 km range. Especially the inversion of the Raman
signals can be distorted dramatically, because the signal slope in the near range changes very much when the
zero-bin for the range correction is varied (see Fig. 1). Hence it is worth some effort to verify that the zero-bin is
really where we assume it to be. In the following we derive the error due to wrong trigger delays analytically in
the next section, and show then how the trigger delay can be determined.

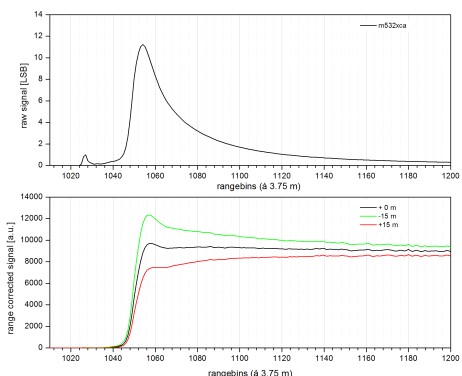

**Figure 1  Top: raw analogue lidar signal with 1026 rangebins pretrigger. Bottom: range corrected signal of the top panel with three different laser trigger delays, i.e.: green, black, and red curves are range corrections assuming that the zero rangebin is at rangebin 1022, 1026, and 1030, which are -15 m, 0 m, and +15 m, from the pretrigger range-bin 1026, respectively.**

### 2.1    Theory

27  The lidar equation for the inelastic Raman backscatter signal is (Ansmann et al. 1992)



$$r^2 P(\lambda_R, r) = C_{\lambda_R} \beta(\lambda_R, r) \exp\left\{ -\int_0^r \left( \alpha_m(\lambda_0, r') + \alpha_p(\lambda_0, r') + \alpha_m(\lambda_R, r') + \alpha_p(\lambda_R, r') \right) dr' \right\}$$ (1)

$P_{\lambda R}$ is the power received from distance $r$ at the Raman wavelength $\lambda_R$, $C_{\lambda R}$ is the lidar constant that contains the
system parameters for this Raman channel, $\beta_{\lambda R}$ is the Raman backscatter coefficient, and $\alpha_m(\lambda_{0,R}, r)$ and $\alpha_p(\lambda_{0,R}, r)$
are the extinction coefficients of air molecules and aerosol particles at the laser and the Raman wavelengths $\lambda_0$
and $\lambda_R$, respectively. For the wavelength dependence of the particle ($p$) and molecular ($m$) extinction coefficients
the Angström-approach (usually with $k = 1$) is used and the parameters $f_p$ and $f_m$ are introduced as:

$$f_p = \frac{\alpha_p(\lambda_0)}{\alpha_p(\lambda_R)} = \left(\frac{\lambda_0}{\lambda_R}\right)^{-k}, \qquad f_m = \frac{\alpha_m(\lambda_0)}{\alpha_m(\lambda_R)} = \left(\frac{\lambda_0}{\lambda_R}\right)^{4.085}$$ (2)

For $\lambda_{\text{Laser}} = 355$ nm, $\lambda_{\text{Raman}} = 387$ nm, and $k = 1$ follows $f_p = 0.917$. Eq. 1 can now be rewritten as

$$r^2 P(\lambda_R, r) = C_{\lambda_R} \beta(\lambda_R, r) \exp\left\{ -\int_0^r \left( \alpha_p(\lambda_0, r')[1 + f_m] + \alpha_m(\lambda_0, r')[1 + f_p] \right) dr' \right\}$$ (3)

The molecular (Rayleigh) extinction coefficient $\alpha_m(\lambda_{0,R}, r)$ can be calculated from actual radio soundings as
shown below in App. 9.2. Assuming that the Raman backscatter coefficient $\beta_{\lambda R}(r) = \text{const}(\lambda_0, \lambda_R) \, \alpha_m(\lambda_0, r)$ is
proportional to the molecular extinction coefficient because both are proportional to the air density, the particle
extinction coefficient $\alpha_p(\lambda_{0,R}, r)$ can be determined from the derivation of the logarithm of Eq. (3) as shown in Eq.

14  (4):

$$\alpha_p(\lambda_0, r) = \left(1 + f_p\right)^{-1} \left( \frac{d}{dr} \ln \alpha_m(\lambda_0, r) - \frac{d}{dr} \ln P(\lambda_R, r) - \frac{d}{dr} \ln r^2 - \alpha_m(\lambda_0, r)(1 + f_m) \right)$$ (4)

A variation/error of the zero-bin by $r_d$ causes a shift of all calculated values including the range correction from
range $r$ to $(r - r_d)$, but not of the measured $P(r)$,

$$\alpha_p(r, r_d) = \left(1 + f_p\right)^{-1} \left( \frac{d}{dr} \ln \alpha_m(r - r_d) - \frac{d}{dr} \ln P(r) - \frac{d}{dr} \ln(r - r_d)^2 - \alpha_m(r - r_d)(1 + f_m) \right)$$ (5)

wherefore we can differentiate Eq. (5) with respect to $r_d$ and see from Eq. (6) that the absolute error of the
particle extinction coefficient due to an error in the zero-bin depends only on $r_d$ and $f_p$ if we neglect the relative
small contributions of the terms containing $\alpha_m$. Below 1 km range this error becomes quite large even for small
zero-bin errors (see Fig. 2).

$$\frac{d}{dr_d} \alpha_p(r, r_d) \approx -\frac{2}{1 + f_p} \frac{1}{(r - r_d)^2}$$ ((6)





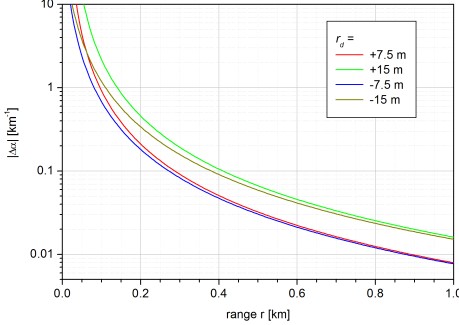

**Figure 2 Absolute error of the extinction coefficient from Raman measurements @355/387 nm due to uncertainties $r_d$**
**of the zero bin according to Eq. 6. Note: as the error is negative for positive $r_d$ the absolute values are plotted to be**
**able to use the log scale.**
Figure 3 shows as an example of the range corrected Raman signals ($\lambda_0$ = 355 nm) measured with the lidar
system POLIS of the Ludwig-Maximilans Universität (Munich) at Praia, Cape Verde Islands on January 29,
2008. The data resolution is 7.5 m. The black line is the range corrected signal using the assumed true trigger
value. The other lines show the range corrected signals with a zero-bin displacements $r_d$ of ±7.5 m and ±15 m.
Below a range of 1.0 km the deviation of the range corrected signals can be seen clearly, increasing with
decreasing range. Figure 4 shows the derived profiles of the particle extinction coefficient with errors due to the
zero-bin uncertainty very close to the theoretically estimated values in Fig. 2.

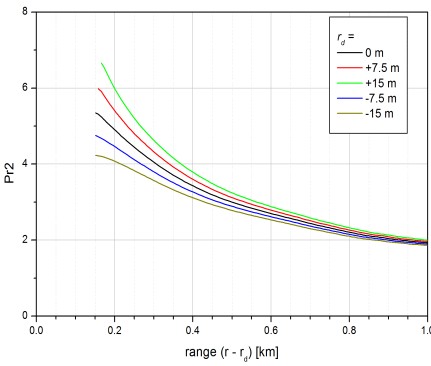

**Figure 3  Range corrected Signal with true zero range  (black) as well as with a zero-bin error $r_d$ of +7.5 m (red), +15**
**m (green), -7.5 m (blue) and -15 m (dark yellow). The Signals have been measured with POLIS on January 29, 2008.**





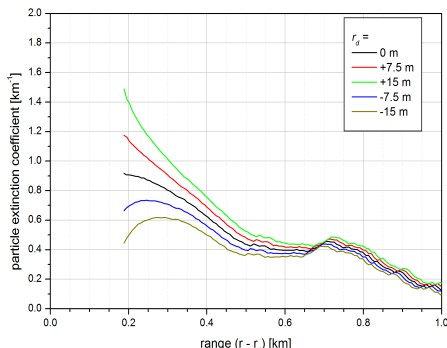

**Figure 4 Same as Fig. 3 but values of the particle extinction coefficient retrieved with the Raman method.**

## 2.2    How to measure the trigger delays

In case pre-trigger samples are recorded, the zero-bin can easily be detected due to the signal peak from stray-light diffusely reflected from the laboratory walls. As the distance to the laboratory walls is not well defined, a diffuse scattering target blocking the laser path (see Fig. 5 top) can be used together with a small hole aperture above the telescope to decrease the signal height to within the detection range of the detectors.

In case no pre-trigger samples are recorded, the zero-bin can be detected by means of a near range target with a known distance to the lidar. Alternatively the sufficiently attenuated outgoing laser pulse can be fed into an optical fibre with sufficient length **s** and the fibre output positioned at the aperture of the telescope (see Fig. 5 bottom). A white open-cell foam often used for instrument packing and a piece of cheap communication fibre (see Fig. 6) served us well for this purpose. With this a signal pulse can be measured with a delay $dt = s / v = s / c * n$ with respect to the outgoing laser pulse, with $c$ = speed of light in vacuum, $v$ = speed of light in the fibre with refractive index $n$ at the wavelength of the receiver channel.

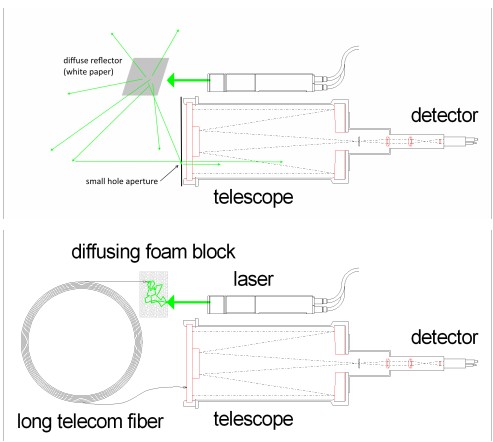

**Figure 5 Lidar trigger delay test setup with diffuse reflector (top) close to the laser and (bottom) with a foam block as beam diffuser/attenuator and a fibre delay line to achieve a controlled trigger pulse delay. The fibre should be as short as possible in order to minimize the wavelength dependent (refractive index) delay error.**





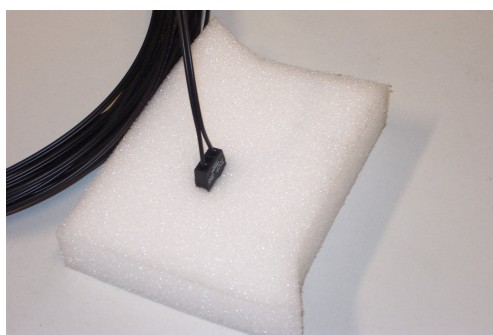

**Figure 6 Open-cell foam and communication fibre used for the zero-bin measurement.**
Note, although we can only measure the range in steps (rangebins) with the resolution of the transient recorder,
i.e. 3.75 m or 7.5 m for the LICEL systems TR40 and TR20, respectively, the uncertainty $r_d$ can take any value as
it results from several electronic delays independent from the transient recorder. Figures 7 shows the analogue
(red/orange) and photon counting (blue/black) signal pulses of a LICEL TR-40 transient recorder with 3.75 m
resolution measured with the setup in Fig. 5 top with about 1000 rangebin pretrigger. The thin lines show
averages of four laser shots and the thick lines the total 88 shot averages. The main peaks of the four shot
averages are distributed between two rangebins. The statistical properties of this distribution can be used to
determine the mean trigger delay and its uncertainty with a resolution better than a rangebin. This is important
for transient recorders with low spatial resolution.

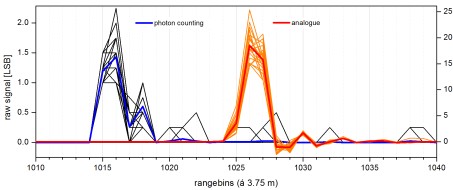

**Figure 7  Trigger-delay / zero-bin pulse of POLIS/Munich 532 nm xp-channel, 25.03.17, four laser shot averages (thin**
**lines) and all 88 shot averages (thick lines) from diffuse reflections off a paper as in setup Fig. 5 top. Black/blue curves**
**are from photon counting with left y-scale and red/orange are analogue signals with right y-scale in LSB. For the**
**channel short-cuts see App. 9.1.**
Figures 7 shows also that the peaks of the analogue signal are about eleven rangebins delayed with respect to the
photon counting (pc) signal (analogue-pc-delay), which is typical for the LICEL transient recorders. Note that
this delay can be different for different LICEL transient recorder modules, wherefore the analogue-pc-delay must
be determined for every module individually. The analogue-pc-delay can also be determined by means of the
cross correlation of the two range corrected signals. Fig. 8 A shows atmospheric signals with the same setup and
at the same date as in Fig. 7, plot B shows the cross correlation of the whole signals and plot C the cross
correlation of the featureless signals between 600 and 1000 rangebins. Both exhibit a distinct peak at −11
rangebins. The small correlation peak stems from the photon noise of the atmospheric backscatter of the laser
pulse, which should be the same in the analogue and in the photon counting signal and detectable in ranges
where the signal noise dominates the background noise.





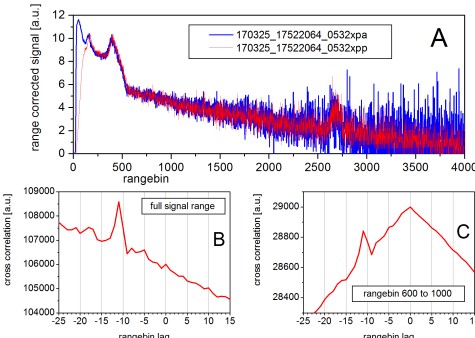

**Figure 8 Relative trigger-delay between the analogue and photon counting signals (A) of a LICEL transient recorder**
**determined from the cross-correlation of signals. Plot B shows the cross correlation of the full signals and plot C of the**
**signal between rangebin 600 and 1000 in a range without aerosol features. Both identify an analogue-pc-delay of**
**eleven rangebins. For the channel short-cuts see App. 9.1.**
The near range peaks of atmospheric lidar signals (see Fig. 9) seem to be unsuitable to determine the zero bin,
which can bee seen in Fig. 10 where the analogue and pc near range peaks of three LICEL TR-40 recorders are
shown (dashed lines) together with the zero-bin peaks (solid lines) measured with a diffuse laser block (Fig. 5
top). The atmospheric near range peaks are one to two rangebins delayed with respect to the zero-bins. Possible
reasons for this delay are multiple scattered light from the near range atmosphere and diffuse reflections from
laboratory walls which are further away. It is remarkable that this delay is less in the cross-polarisation signal,
which should be further investigated.

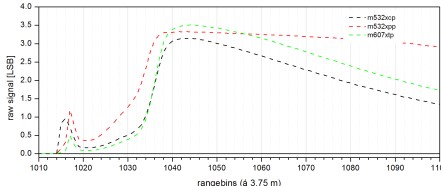

**Figure 9   Raw photon counting signals from normal atmospheric measurements (60k shots averaged) of three**
**POLIS/Munich channels on 27.03.17. For the channel short-cuts see App. 9.1.**

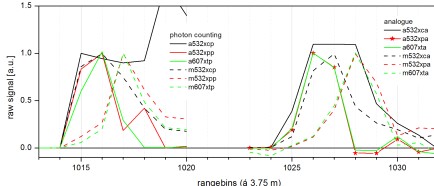

**Figure 10  As Fig. 7 , but with all three LICEL recorder channels, i.e. 532 cross (xc_) and parallel (xp_) and 607 total**
**(xt_) signals. Straight lines (labels a_) show the 88 shot averaged trigger delay measurements as in setup Fig. 5 top,**
**and the dashed lines (labels m_) show the near range peaks from normal atmospheric measurements (60k shots**
**averaged) as in Fig. 9. All signals are normalized to their peak value. The a532xca signal is saturated, and the a532xpa**
**and a607xta partly superpose each other, wherefore the  a532xpa line has additional stars. For the channel short-cuts**
**see App. 9.1.**





**3    Rayleigh fit**
The comparison of lidar signals in clean air ranges with the calculated signals from air density is the only
absolute calibration of lidar signals. To be able to calibrate lidar signals with Rayleigh (molecular) backscatter,
the optoelectronic detection systems must have a high dynamic range. Fig. 11 shows a  POLIS-6 532 nm signal
from a Hamamatsu R7400-P03 photomultiplier recorded with a LICEL TR40-160 transient recorder with 12 bit
A/D resolution and an averaging time of five hours.

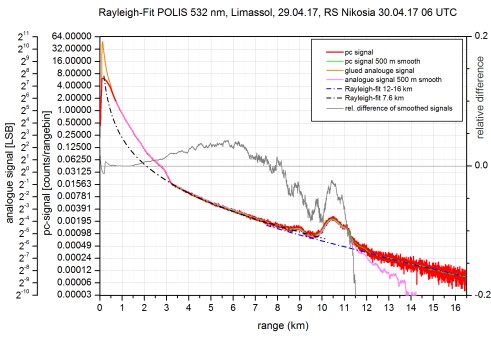

**Figure 11 Dynamic range of a POLIS 532 nm signal (LICEL TR40) averaged over 5 hours and the relative difference**
**between the analogue and the pc-signal.**
The dead-time corrected counts/rangebin of the photon-counting signal (pc, red) is plotted with log2-scale and an
additional y-scale in Least Significant Bits [LSB] to show the corresponding A/D-converter signal level of the
analogue signal. The analogue signal is at the lowest bit limit of the A/D-converter at about 2.8 km range. The
whole glued signal spans a range of about 19 LSB. The comparison of the pc-signal smoothed with a gliding
average of 500 m (green) with the Rayleigh signal calculated from a local radiosonde is shown with Rayleigh-
fits at 7.6 km (black dash-dotted) and between 12 and 16 km (blue dash-dotted). The uncertainty of the fit of the
unsmoothed range corrected signal between 13.3 and 16.3 km (not shown) is less than 1%, i.e. the standard error
of the mean of the residuals relative to the mean Rayleigh backscatter coefficient. This uncertainty is about 4e-6
times the signal maximum in the near range. The relative deviation of the analogue signal, smoothed with a
gliding average of 500 m (light magenta line in Fig. 11), from the smoothed pc-signal is shown as grey line with
the right y-scale. This deviation corresponds to the Rayleigh-fit uncertainty of the analogue signal.
The Rayleigh-fit is a normalization of the range corrected lidar signal to the calculated attenuated molecular
backscatter coefficient ( $\beta_m^{attn}$, Rayleigh signal) in a range where we assume clean air without aerosols and where
the calculated signal fits the lidar signal sufficiently good. Fig. 12 shows an example of an analogue signal where
an aerosol free range is assumed between 5 km and 6 km. Although there are several aerosol signatures below 11
km, the deviation plot at right indicates that the lidar signal can be used up to about 11 km, above which the
analogue signal distortions become too strong.





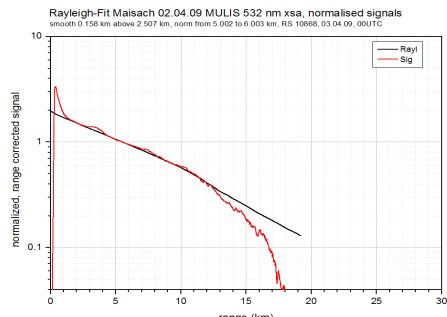
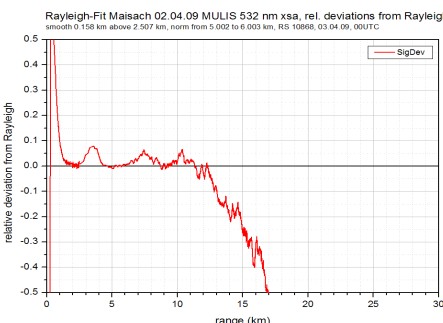

**Figure 12: Left: measured MULIS 532xs (PCI422) analogue lidar signal (red) averaged over 1h and calculated**
**Rayleigh signal (black) from local radiosonde data of the same night, both normalised between 5 and 6 km range.**
**Right: the relative deviation from the calculated Rayleigh signal.**
Because of the signal noise the normalisation of the range corrected signal $r^2 P(r)$ (Eq. (7)) to the calculated,
attenuated molecular backscatter coefficient $\beta_m^{attn}$ has to be performed over a range $(r_{min}, r_{max})$ with the reference
range $r_0$ as centre (Eq. (9)).

$$r^2 P(r) = C\left(\beta_m(r) + \beta_p(r)\right) \exp\left\{-2\int_0^r \left(\alpha_m(r') + \alpha_p(r')\right) dr'\right\} \tag{7}$$

$$\beta_m^{attn}(r) = \beta_p(r) \exp\left\{-2\int_0^r \alpha_m(r') dr'\right\} \tag{8}$$

$$r_{max,min} = r_0 \pm \Delta r, \tag{9}$$

Considering the discrete range-bin resolution this results in the normalized, range corrected signal

$$r^2 P^{norm}(r, r_0) = r^2 P(r, r_0) \frac{\sum_{r_{min}}^{r_{max}} \beta_m^{attn}(r)}{\sum_{r_{min}}^{r_{max}} r^2 P(r)} . \tag{10}$$

Note that for the numerical integration (Fernald-Sassano-Klett inversion etc.) the signal value at the reference
range $r_0$, where the integration starts, must be replaced by the normalized value in order to avoid a noise error,
which means:

$$r^2 P^{norm}(r_0, r_0) = \beta_m^{attn}(r_0) = \beta_m(r_0) . \tag{11}$$

In first approximation, for small aerosol extinction $\alpha_p$, the normalized, range corrected signal of a total
backscatter signal is close to the backscatter ratio around $r_0$ (Eq. 12) and the relative deviation from the Rayleigh
signal can serve as an estimation of the particle backscatter coefficient $\beta_p(r)$ (Eq. 13).

$$\frac{r^2 P^{norm}(r, r_0)}{\beta_m^{attn}(r)} \approx \frac{\beta_m(r) + \beta_p(r)}{\beta_m(r)} \tag{12}$$

$$\frac{r^2 P^{norm}(r, r_0) - \beta_m^{attn}(r)}{\beta_m^{attn}(r)} \approx \frac{\beta_p(r)}{\beta_m(r)} \tag{13}$$



Fig. 13 shows an example of the Rayleigh fit of analogue and pc signals of three LICEL TR80-180 channels, i.e.
the analogue 355 nm cross and parallel polarised signals 355xca and 355xpa and the corresponding pc signals
355xcp and 355xpp, and the Raman scattered signals 387xta and 387xtp. The signal short-cuts are explained in
App. 9.1.

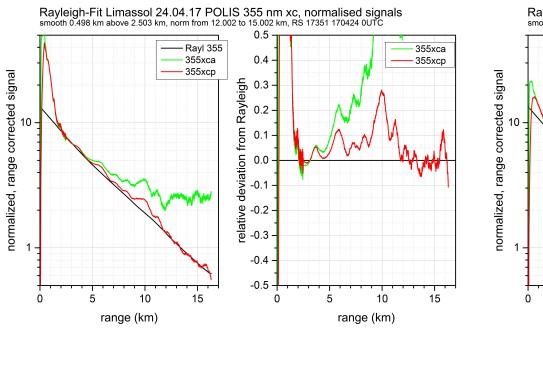

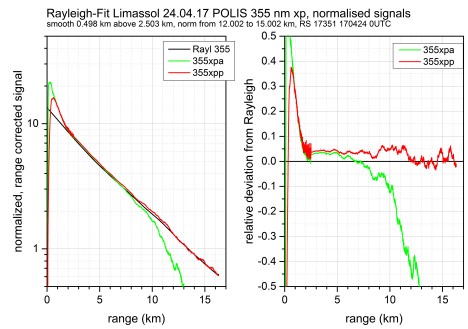

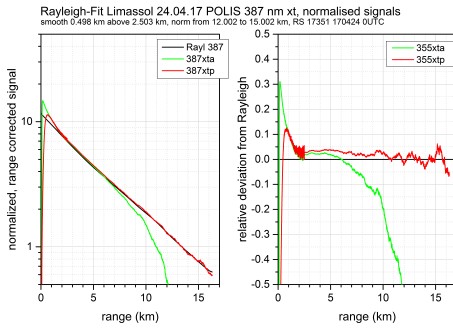

**Figure 13: Rayleigh-fits of POLIS 355xc, 355xp, and 387xt (from top to bottom) analogue (green) and pc signals (red)**
**taken with three LICEL TR80-160. The signals are averages over 2h 45min; left are the signals normalised between 12**
**and 15 km and right are the relative deviations from the calculated Rayleigh signals. For the channel short-cuts see**
**App. 9.1.**
Because the analogue signals of the LICEL transient recorders of POLIS are optimised for the near range, they
start to deviate much earlier from the Rayleigh signal than in Fig. 12 where the MULIS analogue signals are
optimised for the far range. Besides the analogue signal distortions, the Rayleigh-fits and the comparison
between the different wavelengths can reveal errors as, e.g., wrong background subtraction, too high
discriminator level setting of the photon counters (so called hyper-counting), and differences in the receiver
optics.
**4    Lidar test-pulse generators**
Analogue signals suffer from distortions from multiple sources, which cannot be unambiguously recognized and
identified in normal atmospheric lidar signals.




## 4.1    Purpose and description

The accuracy problems of analogue detection channels are clearly visible in section 3 about the Rayleigh-fit above. Signal induced distortions, interspersions of the laser flash lamp and Q-switch triggers and also of the recorder trigger itself as well as the limited bandwidth of the analogue amplifier and its supplementary electronic circuits for range and offset settings etc. are some possible reasons for that. Ground loops in the laser-detector -recorder assembly may add to the signal distortions.

The dynamic range is determined by the ratio of the signal peak value to the amount of noise and signal distortion in the low signal range where the reference value for the Rayleigh calibration is taken. Therefore special emphasis has to be put on low frequency accuracy, i.e. about 30 to 100 µs after the laser pulse (4.5 to 15 km lidar range), where in general Rayleigh calibration is applied.

One of the several sources of errors are the preamplifiers of the A/D converters. Their response to pulses cannot be tested with commercial pulse generators because the accuracy of these pulse generators cannot be guaranteed to the required extent. In turn, the accuracy of the pulse generator can only be determined with the accuracy of the test equipment, i.e. A/D converters. Our solution to this problem was to design an electrical pulse generator with emphasis on a reliable zero signal level after the trailing edge of the test pulse. The original pulse generators (LTPG) had been designed by Dieter Rabus (MIM) , improved by the *Institute for Tropospheric Research* (IFT), and finally re-designed and three units (LISIG1,2,3, see Fig. 14) distributed to EARLINET groups by Holger Linné (MPI) for testing various analogue recorders.

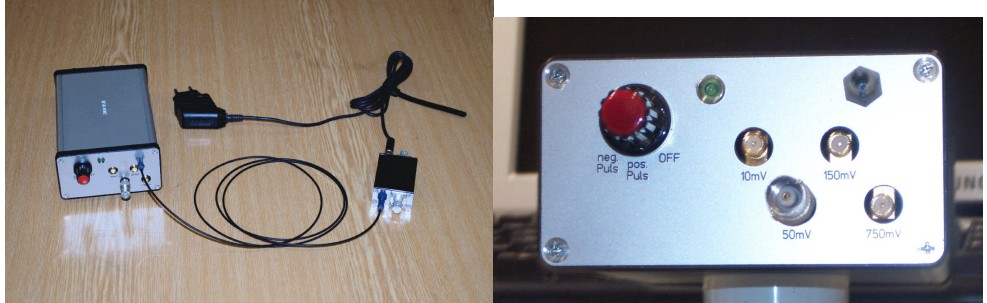

**Figure 14  Lidar test pulse generator LISIG2 with opto-decoupled trigger generator and fibre connection**

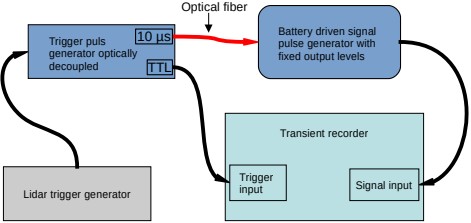

**Figure 15   Setup of the lidar test pulse generator and the opto-decoupled trigger generator (LISIG , blue at top) between the lidar systems tirgger generator (grey) and the transient recorder with the A/D converter (light blue).**



The LISIG test pulse generators produce a selectable negative or positive square pulse with 10 µs length,
corresponding to a boundary layer with 1.5 km top, and fixed pulse heights of nominal 10 mV, 50 mV, 150 mV,
and 750 mV, with the base at 0 V. The corresponding measured output voltages of LISIG2 with new batteries are
8.8 mV, 44 mV, 148 mV, and 616 mV. The emphasis of these pulse generators is not on the perfect square pulse
shape and rise/fall times, but on the reliable, return to ground level after about 30 µs with subsequent low
frequency ripples well below 10 µV amplitude. To avoid ground loops as well as cross talking of digital
electronics into the analogue data line, the LISIG is split into two separate devices. The first device generates a
rectangular pulse of approximately 10 µs length. For this a simple integrated circuit of type 555 is used. Its
output is coupled to a separate analogue electronics box by using an optical link based on a plastic fiber system.
The analogue electronics only contains a fast RF switch (ZYSWA-2-50DR by Mini-Circuits) that switches on
and off a battery voltage on the analogue signal output according to the optical input signal. This way, the signal
is guaranteed to return to zero (the negative pole of the battery) in a few nanoseconds. The analogue electronics
box has an electrical connection to the A/D converter only through the analogue signal cable. To ensure this, the
analogue electronics are placed in an isolating plastic box. Therefore the LISIG can be placed in the real
measuring environment simply by replacing the detector.
**4.2    Measurements with the test pulse generators**
Within EARLINET many different transient recorders have been tested with the LISIG. As an example how the
impulse response can be improved and what can be achieved we present the tests of three different transient
recorders: i.e. a SPECTRUM PCI412-40  4-channel transient recorder with 40MHz and 12 Bit resolution, a
SPECTRUM MI4022-20  4-channel transient recorder with 20MHz and 14 Bit resolution, and three LICEL
TR20-160 version 2007 with 20 MHz. The responses to the test pulses at different range settings are shown in
Figs. 16 and 17.

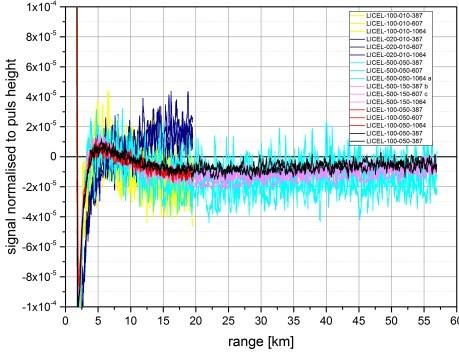

**Figure 16  Pulse responses of three LICEL TR20-160 transient recorders at three range settings to square test pulses**
**with 10, 50, and 150 mV height and 10 µs width. The labels LICEL-A-B-C show A: range setting in mV, B: pulse**
**height in mV, C: recorder channel (387 and 607 and 1064 nm channels of MULIS). The x-scale is lidar range, i.e. 10**
**km correspond to 150 µs record length. The signals are recorded with 20 MS/s and are smoothed with a running**
**average of 25 range bins corresponding to 187.5 m. Some signals were only recorded up to 20 km range.**



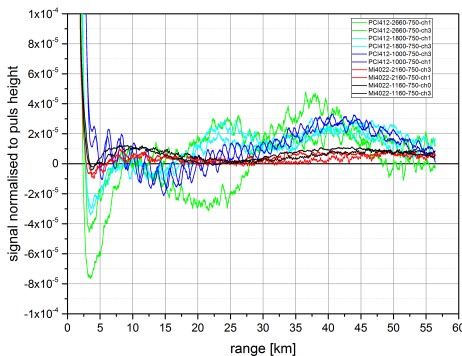

**Figure 17  Pulse responses of several channels of two SPECTRUM transient recorders (PCI412 and MI4022) at**
**different range settings to square test pulses from LISIG2 with 750 mV height and 10 µs width. The labels X-A-B-C**
**show X: recorder type, A: range setting in mV, B: pulse height in mV, C: recorder channel. The x-scale is lidar range,**
**i.e. 10 km correspond to 150 µs record length. The signals were recorded with 20 MS/s and are smoothed with a**
**running average of 67 range bins corresponding to 502.5 m.**
While the pulse response of the PCI412 is not bad for a commercial recorder, the MI4022 responses are much
improved with a distortion to pulse height ratio of 1e-5. As a first step of optimisation all electronic input circuits
of the preamplifiers of both recorders had been removed by the manufacturer except the electronic switch for
three fixed range settings. The further improvement in the MI4022 has been achieved by individual tuning of the
amplifier input stages of each channel using an earlier version of the test pulse generator as a reference.
Further results of our test measurements are that the distortions have a component which increases about linearly
with the pulse length, and together with the shown increase with pulse height shown in Fig. 16 we can see an
about linear increase with pulse area. We also saw considerable cross talk between the channels of the MI4022,
which might be present also in other models. Such cross-talks for example between cross-polarised signal
channels and the Raman- or parallel-polarised channels can cause considerable signal distortions in ranges with
high depolarisation ratio. The results for the presented LICEL transient recorders are typical for all the several
other LICEL transient recorders tested by several EARLINET groups.
**4.3    Linearity of the analogue output of MB-01/02 photo-receiving modules with respect to the optical**
**input**
LISIG tests only the electronic part of the photo receiver. It cannot be used to test the integrated photo-receiving
modules MB-01 and MB-02 (Fig. 18), which are widely used in the lidar systems of CIS-LiNet, because the
photo-detector and the whole A/D conversion electronics as well as the amplification and power supply and
transient recorder are included in one module in order to avoid ground loops and external EM-interspersions.





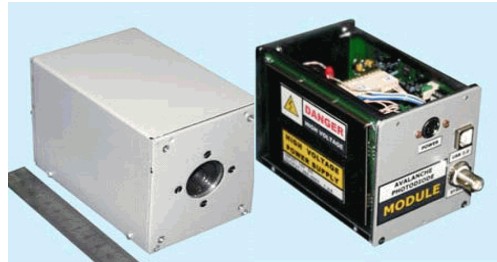

**Figure 18: Integrated photo-receiving modules MB-01/02 of CIS-LiNet lidar systems, including either C30956E-TC APDs or FEU-85 / FEU-175 PMTs and the whole power supply and A/D conversion and amplification electronics.**

The analogue pulse height linearity of these modules has been tested with a pulsed, external light source as shown in Fig. 19. Figure 20 shows the normalised output/input pulse height ratio of an MB-01 module with an FEU-84 photomultiplier at F-in levels spanning four decades with two different background light levels. The measurements show that the output/input ratio non-linearity is less than ±2% over four decades of input level.

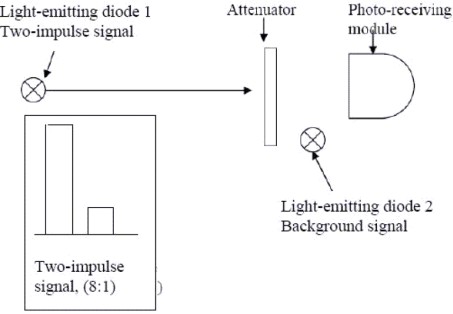

**Figure 19: Setup for testing the pulse-height linearity of the integrated photo-receiving modules MB-01 / MB-02 with two external LEDs, one pulsed with two consecutive square pulses with pulse heights of 8:1 ratio and different optical attenuation, and the other as constant background light source.**

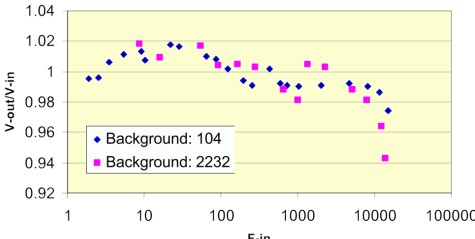

**Figure 20: Normalised output/input pulse height ratio of an MB-01/FEU-84 module at changing F-in levels and two background light levels.**

## 5    Dark measurement

If signal distortions are independent of the lidar signal, they can be determined with so-called dark-measurement.

The measured dark-signals without atmospheric backscatter can be subtracted from the normal lidar signals just as the skylight background or the analogue DC-offset, but as range dependent offset.





The dark measurement is like a normal measurement with laser and Q-switch trigger etc., but with fully covered
telescope, so that no light from the atmosphere and from the backscattered laser pulse is collected by the
detectors. In such signals we can see EM-interferences from the electro-magnetic laser pulses or other electronic
interferences which are synchronous to the laser trigger, but also rests of analogue low frequency noise, which
can never be completely removed by means of spatial or temporal averaging (see Fig. 21 C). As there are
different sources of such disturbances with different effects on averaged lidar signals, we currently don't have a
standardised procedure for the dark measurements and cannot use them for the evaluation of the lidar signal
quality in a standardised way. However, if after sufficient temporal averaging of the dark measurement the signal
distortions are stable, which means not changing by further temporal averaging, the dark signals can be
subtracted from the atmospheric signals to improve their accuracy. Figure 21 A shows an example of a 1064 nm
LICEL APD analogue signal where the subtraction of a dark signal from the raw signal could be applied, which
is verified by a Rayleigh fit and the Klett inversion backward and forward from the fitting range, chosen between
5 to 10 km (Fig. 21 B). Figure 21 C shows the unsmoothed lidar and dark signals in the near range, where strong
high frequency interspersions, probably stemming from a trigger signal, are visible at about 1.4 km
corresponding to about 10μs. Because it is not practical to make the dark measurements for a timespan
comparable to the atmospheric measurements (1:15 h in Fig. 21), the subtraction of the dark measurement with
the same smoothing length as the atmospheric measurement (Fig. 21 A, blue line) would considerably increase
the signal noise in the far range. On the other hand, with a high dark signal smoothing in the near range the high
frequency interspersions (Fig. 21 C) could not be removed. We therefore recommend to not smooth the dark
signal in the near range and to start smoothing only when it would increase the signal noise. Furthermore, we
found that the near range interspersions can change quite fast. Hence it is necessary for each channel to test the
temporal stability of the dark signal regularly before using it for signal correction.

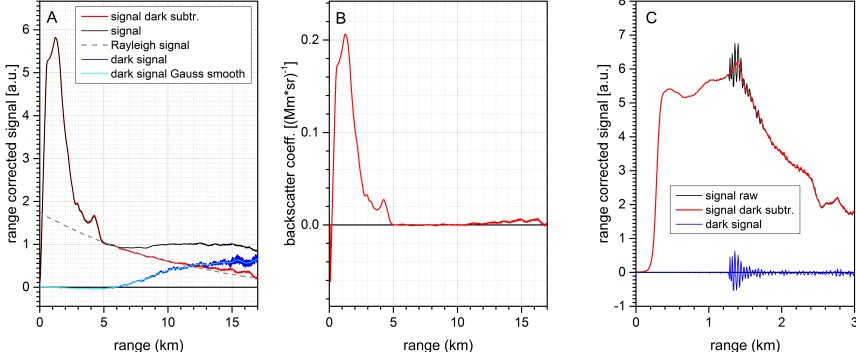

**Figure 21: 1064 nm lidar signal with a LICEL 3 mm APD (black) with a LICEL TR20 transient recorder averaged**
**over 1h 15min, dark measurement (blue) 8 min average, and corrected lidar signal (red) together with the calculated**
**Rayleigh signal (grey dashed). Plot A and B show sliding averages over 500 (cyan line: sliding Gaussian smooth with**
**sigma 500 m), while in plot C the original resolutions are shown.**


## 6   Telecover test
Deviations of the near range signals from different parts of the telescope and the comparison of such deviations
of different lidar channels and with theoretical ray-tracing simulations can reveal the distance of full overlap and
possible reasons for the deviations from the ideal case.
In contrast to the far range, where we can use the Rayleigh-fit in clear air ranges, we don't have a calibration
method for a lidar system in the near range, where almost never clean air conditions can be assumed. But
shortcomings of the optical and opto-mechanical design or misalignments have their largest effect in the near
range. A test for this range is based on the fact that the backscattered photons collected by different parts of the
telescope of a lidar system must give the same range dependency of the signal, and if not, the range dependency
of the whole signal is uncertain. With ray tracing simulations we see that ray bundles collected by different
telescope parts reach the signal detector in different paths through the optical receiver and hit the optical
components under different incident angles (see Fig. 22), with possibly different transmission. Possible causes
for the differences are laser tilt, telescope misalignments, displacement of field and aperture stops (vignetting,
defocus), optical coating effects of, e.g., beam-splitters and interference filters with spatial inhomogeneity or
angle dependency of the transmission (see Fig. 23), or spatial inhomogeneity of the detector sensitivity
(Simeonov et al. 1999). The geometrical overlap function, which is mainly determined by the size and location
of the telescope's field stop, is just the most obvious feature producing differences in different telecover signals.

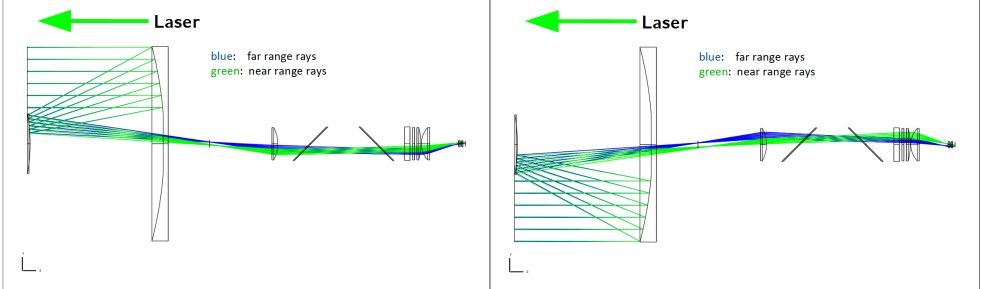

**Figure 22: Ray bundles through the receiver optics of a typical lidar setup from the top part of the telescope (left) and**
**from the bottom part (right), from near range (green) and far range (blue) have different paths and incidence angles**
**on the optical elements.**

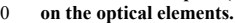
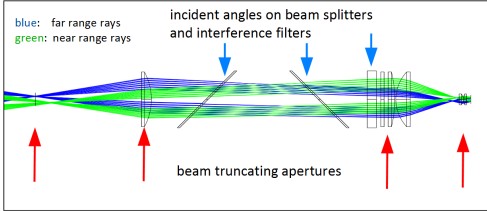

**Figure 23: Optical elements in a typical lidar receiver optics which can influence the transmission of the ray bundles**
**due to vignetting (red arrows) or angular transmission dependency (blue arrows).**
In a first attempt the telescope can be covered in a way that just quarters of the telescope are used, which we call
the Quadrant-test (see Fig. 24), or using only an inner and outer ring of the telescope, i.e. the In-Out-test. Using
In-Out sections of the quadrants is called the Octant-test.

1                                                                                                                   16





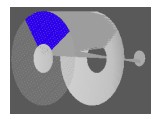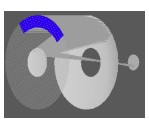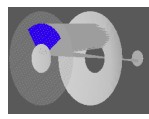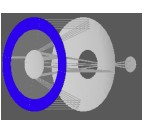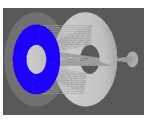

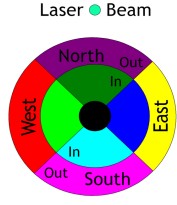

**Figure 24: Nomenclature of the telecover parts (plot at right) with respect to the laser position at North (biaxial systems) or any prominent orientation of the receiver optics (mono-axial systems). Using the four quarters N,E,S, and W in the left picture is called the quadrant test. Using the outer and inner parts of the quadrants is called the octant test. The pictures above show (from left to right) the sectors North (N), North-Out (NO), North-In (NI), Full-Out (FO), and Full-In (FI) on a telescope, assuming the laser on top.**

With an ideal lidar system the normalized signals from all different telecover tests must match - apart from the overlap range, which can be therewith assessed, and assuming constant atmospheric conditions during the test. Figure 25 shows an example of quadrant telecover signals from three POLIS-6 channels. The cyan N2 signals are taken with the same telecover sector as the N signals, but at the end of the temporal sequence of the measurements. Deviations between N and N2 signals indicate the influence of the changing atmosphere during the measurements, which are here visible in the cross polarised 355xcg signals between about 200 m and 500 m range, but much less pronounced in the other channels. The relative differences between the normalised signals are well below 5% in the near range and mainly due to signal noise, except for atmospheric disturbances.

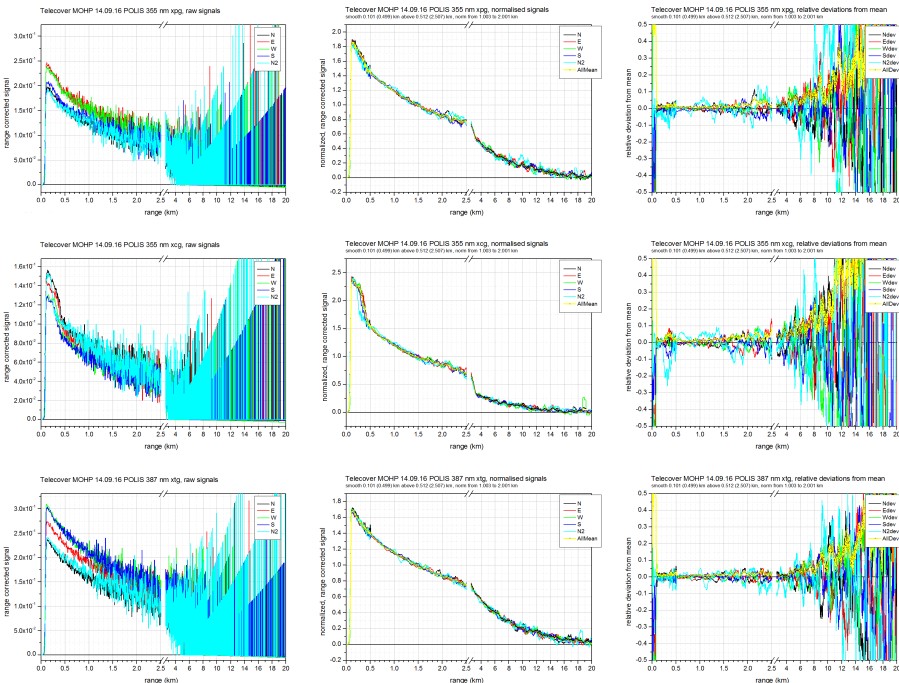

**Figure 25: Quadrant telecover signals from three POLIS-6 channel: 35xcp, 355xpp, and 387xta (see App. 9.1 for shot-cuts). The left plots show the raw, range corrected signals, the middle plots the smoothed and normalized signals, and the right plots the relative deviations from the mean of all signals but N2.**



The distance of full overlap is not below about 100 m. In this case of a well aligned lidar system with well
designed eyepieces, the raw signal differences between the telecover sectors indicate just the different
sensitivities of the different areas of the photomultipliers, which are different in the three channels. The
measured telecover differences can be compared to paraxial (see also Kokkalis (2017)) and exact ray tracing
(ZEMAX) simulations of the system including apertures and optical coatings to narrow down possible causes.
Figure 26 shows the paraxial near range simulation of the field of view for the full telescope aperture (left), and
for the N (blue) and S (magenta) telecover sector with a circular field stop in the focal plane of the telescope
(neglecting the obscuration of the secondary mirror). The distance of full overlap between the full telescope and
the laser (grey) is only reached when the laser beam is fully in the light green inner core of the field of view of
the full telescope. The distance of the full overlap is reached earlier for the N sector than for the S sector (Fig.
26, right). This difference is very small for the POLIS-6 in Fig. 25, because there a tilted slit field stop is used
(Freudenthaler 2003).

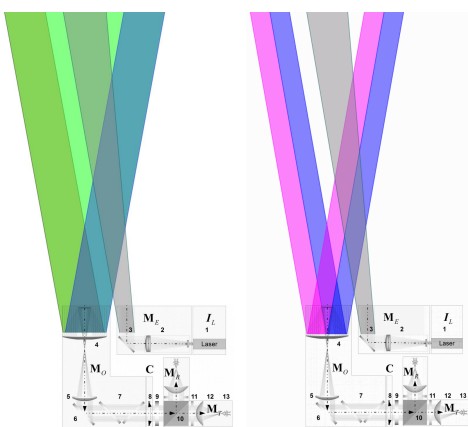

**Figure 26: Schematic field-of-view (paraxial simulation) of a two-channel lidar system with a laser with beam**
**expander and steering mirror, a telescope, and the receiving optics. The laser beam (grey) with a small divergence is**
**tilted towards the axis of the telescope. The left plot shows schematically the limits of the field of view of the full**
**telescope and their overlap with the laser beam. The full overlap of the laser beam and the filed of view of the**
**telescope is only reached when the laser beam is fully in the inner light green core of the field of view. The right plot**
**shows the same but for the N (blue) and S (magenta) sector of the telescope. The full overlap with the N sector of the**
**telescope is reached earlier than with the S sector.**
Figure 27 shows the paraxial simulations as in Fig. 26 but for different defoci of the field stop and for the far
range (top rows) and the near range (lower rows). A negative/positive defocus results in a later/earlier distance of
full overlap and generally in a loss of full overlap in the far range. Comparing the telecover simulations in the
near range (Fig. 27, lowest row), we see that at +20 mm defocus the distance of full overlap is reached earlier for
the S than for the N sector, which is in contrast to the situation with defoci smaller than about +10 mm. In the far
range the full overlap is earlier lost for the N sector than for the S for negative defoci and vice versa for positive
defoci.





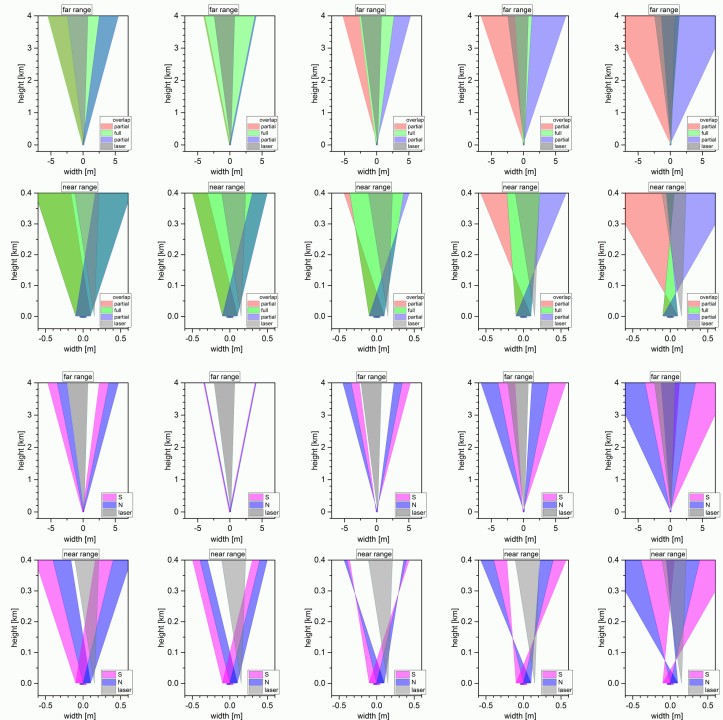

**Figure 27: Same as Fig. 26, but with different defoci of the field stop (from left to right): -5 mm, 0 mm, +5 mm, +10 mm, and +20 mm. The telescope parameters are f = 1200 mm, diam. 200 mm, radius of the circular field stop 1.2 mm; laser-telescope axes distance 150 mm, and laser divergence fw 0.8 mrad .**

An exact ray-tracing simulation (ZEMAX) of the relative telecover deviations in Figs. 26 and 27 is shown in Fig. 28 for defoci of 0 mm (top) and +10 mm (bottom) with a Gaussian laser beam with full width divergence of 1 mrad encircling 86% of the energy. As expected from the paraxial simulation in Fig. 27, the decrease of the distance of full overlap from about 300 m to about 150 m between 0 and +10 mm defocus can be identified while closing the gap between the N and S-signal. There is also the concurrent loss of full overlap of the S-signal in the far range with deviations in the 5% range, which are almost hidden by the signal noise and due to the noise in the normalisation range (2 to 4 km). But it shows that already relative deviations in the 5% range reveal a critical situation of the optical setup, wherefore the measured signals should have an as good as possible signal to noise ratio.



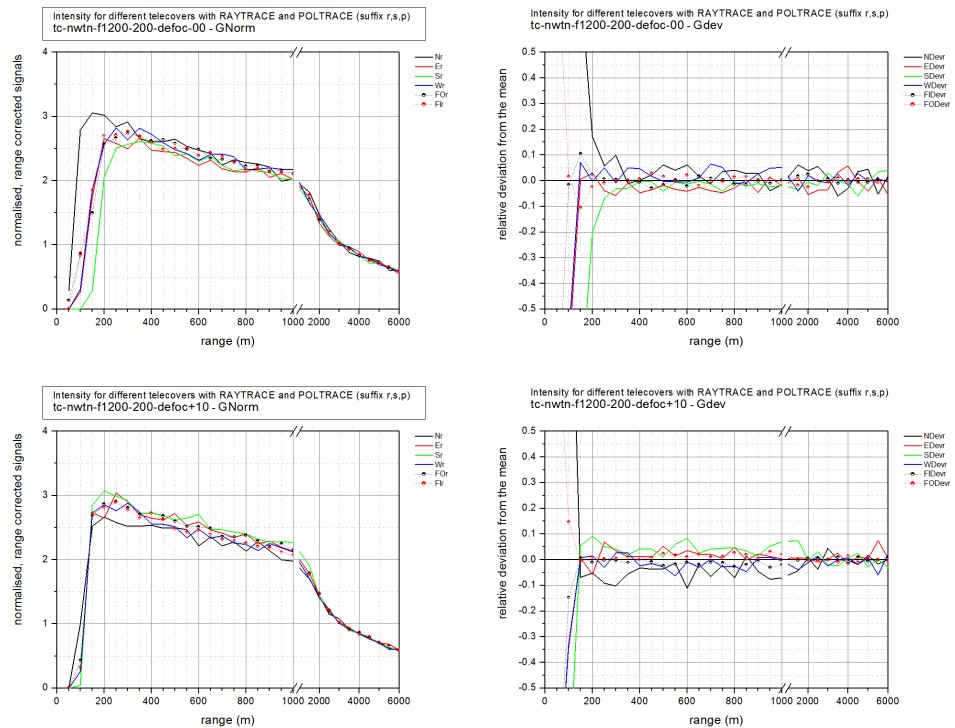

**Figure 28: Exact Monte-Carlo ray-tracing simulations of the normalised telecover signals NESW, FI and FO with 5000 rays considering polarisation and the relative deviations for the cases of Figs. 26 and 27 for 0 mm (top) and +10 mm (bottom) defoci.**

In Fig. 29 we see a disadvantage of the telecover test. It shows the effect of a S-N tilted, small bandwidth interference filter together with a laser S-tilt of 0.25 mrad. At +1.5° filter tilt (bottom) the relative deviations from the mean (right) are clearly visible even above 3 km, but at -1.5° filter tilt (top) the relative deviations from the mean seem to vanish above about 600 m - in contrast to the deviations of the normalised signals, which differ up to at least 1 km from the signals without filter consideration (lines with open circles). But this information we have only in the simulations, not in the real world. This shows that if the misalignments/distortions of the lidar setup affect all telecover signals in a similar way, the relative deviations can't show it, and in the normalised signals we can't distinguish such distortions from the aerosol signature in the near range. Therefore we must consider a perfect telecover test as a sine qua non and not as a sufficient condition for an ideal lidar setup. However, the difference between Fig. 29 top and bottom shows clearly the combined effect of the interference filter and laser tilts. Considering the collimator lens (see Fig. 23) as a Fourier-transform lens for the angular transmission function of the interference filter, the latter can be transformed in a spatial transmission function at the location of the lens' focal plane, i.e. the location of the telescope's field stop, and the combined overlap function of the receiver optics is the convolution of interference filters' and the field stop's spatial transmission functions.





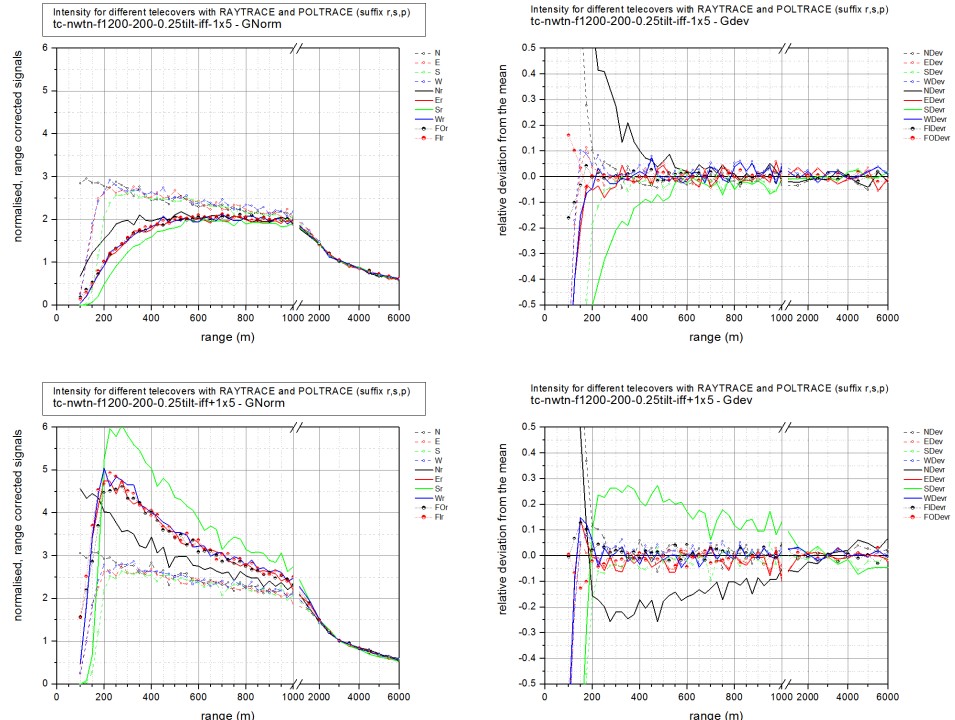

**Figure 29: As Fig. 28 for 0 mm defocus but with a laser tilt of 0.25 mrad towards the telescope and a small bandwidth**
**interference filter in the parallel beam path (see Fig. 23) with 0.19 nm bandwidth and 1.8° acceptance angle at 90% of**
**maximum transmission (coating 532-IFF2C5555HISM20 ). The interference filter behind a paraxial collimator with f**
**= 70 mm is tilted -1.5° (top) and +1.5° (bottom) in S-N direction. The signals without appendix "r" (open circles) are**
**for comparison and calculated without the interference filter.**
As the incidence angles at the telescope aperture are magnified by the telescope-collimator combination by a
factor of $(-\ f\_telescope\ /\ f\_collimator) = -1200\ /\ 70 = -17$, the +0.25 mrad S-tilt of the laser is equivalent to a –
17 x 0.25 mrad = –4.25 mrad = –0.25° tilt of the interference filter. Furthermore, a difference between the laser
wavelength $\lambda$ and the centre wavelength of the interference filter $\lambda_0$ causes the same effect as a difference
between the laser and interference filter tilts ($\alpha$) according to the relation between the centre wavelength shift
and incidence angle in the interference filter (Eq. (14)).
$$\lambda = \lambda_0 \sqrt{1 - \left(\frac{\sin\alpha}{n}\right)^2} \Rightarrow \left(\frac{\sin\alpha}{n}\right)^2 = \left(\frac{\lambda_0 - \lambda}{\lambda_0}\right) \tag{14}$$
A simulation example for a coaxial lidar (same as above but without laser-telescope axes distance and
interference filter) is shown in Fig. (30) with a defocus of -10 mm (top) and an additional laser tilt of 0.5 mrad
(bottom). In the first case, without laser tilt, the Quadrant test (NESW) doesn't show any deviations, but the In-
Out test does up to about 500 m. In contrast, with laser tilt the In-Out test deviations are weak above 300 m but
the Quadrant test shows considerable deviations up to 2 km. Especially coaxial lidar systems should always be
tested with both, the Quadrant and the In-Out tests.





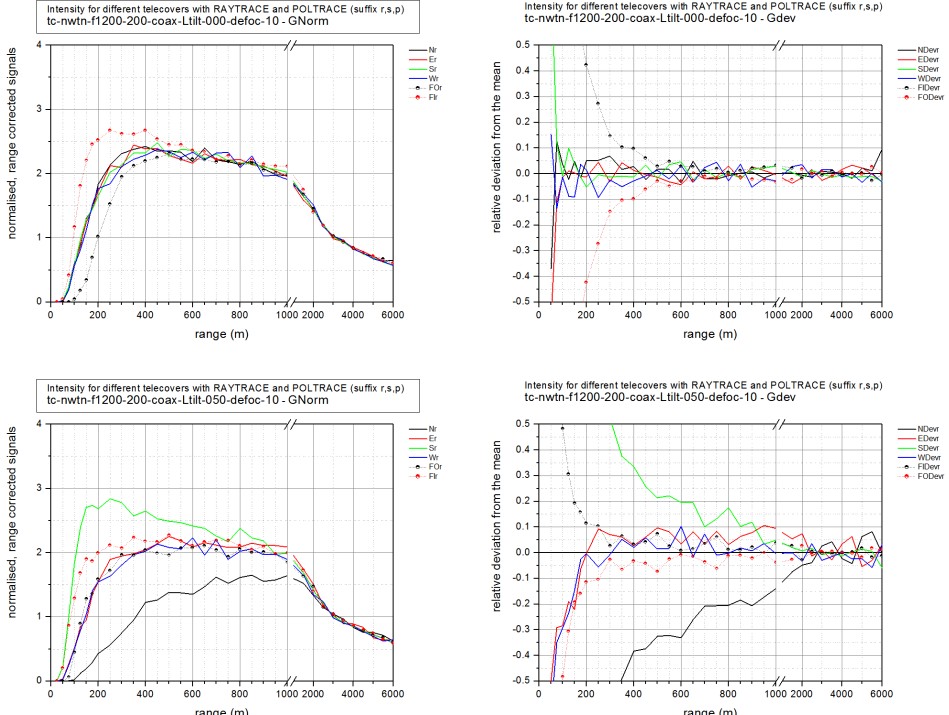

**Figure 30: Exact Monte-Carlo ray-tracing simulations of the normalised telecover signals NESW, FI and FO with**
**5000 rays considering polarisation for the same lidar setup as in Fig. 27 - but coaxial and the relative deviations from**
**the mean for -10 mm defocus and a laser tilt of 0 mrad (top) and 0.5 mrad (bottom).**
Further examples of what the telecover test can reveal are shown in Figs. 31 and 33. In Fig. 31 the Quadrant
telecover test signals of two channels of a lidar are plotted, with very different near-range deviations in the two
channels. The small Hamamatsu photomultiplier R5600 and its successors as the R7400 exhibit a strong
inhomogeneity of the detection sensitivity across the detector surface as shown in Fig. 32 plot A and B
(Simeonov et al. 1999).

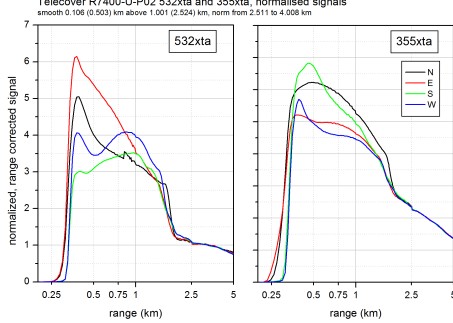

**Figure 31: Quadrant telecover test of two channels of a lidar with Hamamatsu R7400 photomultipliers. The deviations**
**between the sector signals are very different for the two channels.**



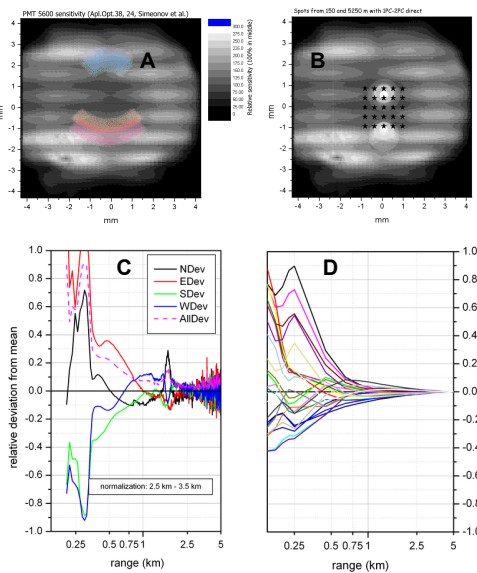

**Figure 32:** The telecover test of another lidar system shows irregular deviations in the near range (plot C) of all sectors, which can be explained by the inhomogeneity of the photomultiplier sensitivity as shown in the plots A and B for a Hamamatsu 5600 PMT (Simeonov et al. 1999) and the movement of the laser spot over the detector surface with the change of the lidar range. The movement of the NI- (blue) and SO-sectors (red) over different sensitive parts of the PMT are indicated in image A for 0.25 and 5 km lidar range. Plot D shows the possible signal deviations from a set of simulations with various PMT alignments as indicated by the starts in plot B (Freudenthaler 2004), which are very similar to the measurements. An additional eyepiece in front of the PMT could possibly solve this problem.

In case the laser beam is focused on the PMT, the laser spot moves over the detector area with the lidar range, especially in the near range, and the measured signal reflects rather the detector inhomogeneity than the atmospheric structure. A simulation of the possible relative deviations from the mean of the Quadrant signals (Freudenthaler 2004)is shown in Figure 32 D together with the measured ones in plot C. Plot A and B show the simulation conditions for the location and movement of the beam spot on the PMT. A possibility to countercheck the influence of the PMT is to rotate the PMT by 90° and to compare the signal features of both Quadrant tests.

The Quadrant measurements of MULIS in 2003 in Figure 33 (left) show a loss of signal intensity of the N-signal in the far range. The first assumption would be a misalignment of the laser with a S-tilt (compare Fig. 27 for -5 and 0 mm defocus), but the comparison with the signal simulations (right) with a good agreement in the onset of the overlap indicated no misalignments of the laser, defocus of the telescope, or tilt of the interference filter. The second signal from the E-sector (E2, cyan) coincides with the first signal (E, red) and shows that atmospheric changes didn't influence the test. A closer inspection of the receiver optics by means of a CCD-camera was done, and looking through the receiver optics at an image of the telescope aperture we saw strong distortions in the N-sector of the telescope (at NW in the right image plot of Fig. 33), probably due to stress in the thin secondary mirror of the Cassegrain telescope. This lead to the decrease of signal intensity from the near to the far range in the N-signal. The preliminary solution was to mask the N-sector permanently, and the final solution to replace the telescope.





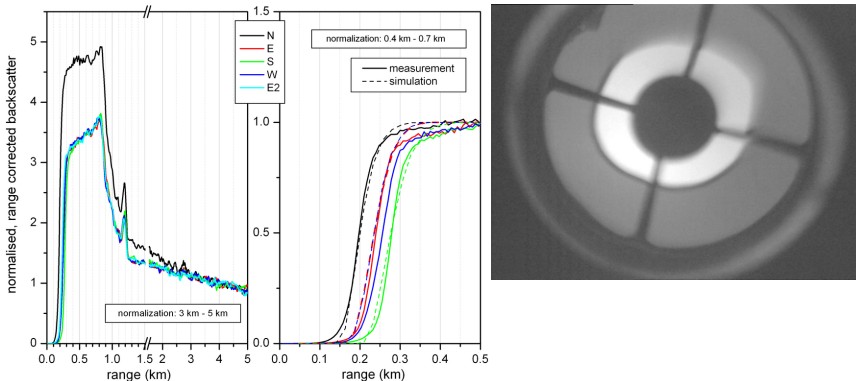

**Figure 33: The left plot shows normalized telecover measurements with a strong deviation of the N-signal from the other sectors. But normalizing the same signals in the near range, the mid plot shows a very good agreement between the measured and the simulated overlap regions for all sectors. Considering all additional information, the telescope had been inspected. The right plot shows an image of the telescope aperture through the receiver optics, which indeed shows enhanced image distortions in the N-sector, probably due to stress of the secondary mirror.**

## 7    Polarization calibration with error analysis

For the calibration of the relative sensitivity of the two polarisation channels and for the compensation of the effect of polarizing optical elements in lidar systems the EARLINET QA requires a Δ90 calibration (or similar) and a comparison of the corrected linear depolarisation ratio (LDR) in clean air ranges with the calculated molecular LDR – similar to the Rayleigh calibration in section 3. The theoretical background of the Δ90-calibration is described in detail in Freudenthaler (2016a). Figure 34 shows the corresponding measurements of POLIS-6 during an episode with rather clean atmosphere and their analysis. The top left plot with a log-y-scale shows the range corrected and smoothed calibration signals (IT for transmitted and IR for reflected intensities behind the polarising beam splitter) at 532 nm with a rotated calibrator, which is in this case a mechanical rotation of the whole receiver optics behind the telescope (Freudenthaler et al. 2015) (Freudenthaler 2016a; section 7.2) . Also shown are the normal so-called Rayleigh signals (the transmitted ITRayleigh and the reflected IRRayleigh), which should contain ranges without aerosol for the comparison with the calculated molecular depolarisation ratio. While the calibration should be done as often as possible, also during daylight if necessary, and therefore cannot be too long, the Rayleigh measurement should be done at night and as long as possible to decrease the signal to noise ratio in the aerosol-free region. They don't have to be at the same day or night, but with the same system settings as the calibration measurements. However, we recommend to do the Rayleigh measurement as close as possible to the calibration and if possible without switching off the lidar or the data acquisition, because already small changes in the PMT high voltage supply or of the PMTs themselves due to environmental or system temperature or temperature changes of the second and third harmonic crystals of the laser can influence the accuracy. Even the data acquisition electronics doesn't necessarily settle in the same state at each power-on.

The measurements in Fig. 34 were done at the same night (15.09.16, 20:38) as the calibration measurements (16:59). The following references *F16* refer to Freudenthaler (2016a). The top, right plot in Fig. 34 shows the range dependence of the calculated calibration factors at the ±45° positions and the geometric mean using Eqs.



(80), (84), and (85) in *F16*, section 5, with Vplus = $\eta(+45°)$, Vminus = $\eta(-45°)$, and V = $\eta_{\Delta90}$ , as well as an
estimation of the misalignment of the calibrator rotation $\varepsilon$ (*F16*, section 11) in the header with the uncertainty of
$\varepsilon$ and $\eta$ due to signal noise. The broken lines Vplus_mean and Vminus_mean indicate the calibration range and
the mean values. The insert lists the GHK-parameters (GR, GT, HR, HT, K) (*F16*; sections 4 .1 and 5), the
rotation of the plane of polarisation of the laser (RotL = $\alpha$), the system orientation with respect to the reference
plane of the optics (y = 1), and the factor NDfac by which the cross signal is attenuated during the calibration
measurements by means of an neutral density filter.
The GHK parameters and the resulting systematic uncertainties of the LDR for different lidar systems and
calibration techniques can be calculated using the corresponding equations in *F16* and the numerical technique
described in Bravo-Aranda et al. (2016) and by means of the open source Python script (Freudenthaler 2016b),
which uses these equations.
With these parameters the uncorrected linear depolarisation ratio of the Rayleigh measurement (LDRmeas = $\delta^*$;
red line) in Fig. 34 (bottom, left) is corrected (LDRcorr = $\delta$; green line) according to *F16*, Eqs. (60) and (62).
The blue line LDRcorr_mean indicates the range where the signal is assumed to be aerosol free and with
sufficient SNR to calculate a mean value of the smallest LDRmeas and LDRcorr in the signal and their
uncertainties due to noise, which are both mentioned in the insert. The right bottom plot shows the same as the
left, but for 355 nm, The signals and calibration are not shown for this wavelength. The atmosphere at this night
was indeed very clean down to about 3 km, which enabled us to use a quite low Rayleigh range between 3.5 and
5.1 km, in which the LDRcorr are $0.0103 \pm 0.0001$ at 355 nm and $0.0055 \pm 0.0001$ at 532 nm.
We can compare these values with the ones measured in 2013 and the expected Rayleigh LDRmol
(Freudenthaler et al. 2015), i.e. LDRmeas = $0.00824 \pm 0.00021$ at 355 nm and $0.00546 \pm 0.00031$ at 532 nm,
LDRmol = $0.00785 \pm 0.00024$ at 355 nm and $0.00444 \pm 0.00008$ at 532 nm. It must be mentioned that the
uncertainties of the 2013 values include the systematic uncertainties and the variation over about one month,
while the 2016 uncertaities only contain the signal noise. The 532 nm value is very close to the expected and the
2013 values, showing that the atmosphere was really clean, but the 355 nm value is significant higher with
LDRcorr = 0.0103 instead of 0.00824 (2013) and 0.00785 (Rayleigh). The reason is probably the change of the
laser in 2015 with different SGH and THG crystals and laser output window.
The prospective plan for the polarisation calibration test is to combine in the analysis the random and the
systematic uncertainties determined with the Python script. While the calculation of the calibration factor and the
GHK-corrections as shown above are already installed in the Single Calculus Chain (SCC, D'Amico et al.
(2015); D'Amico et al. (2016)), which is the common EARLINET lidar signal analysis sofware, also here the
combined consideration of random and systematic uncertainties is a future task.





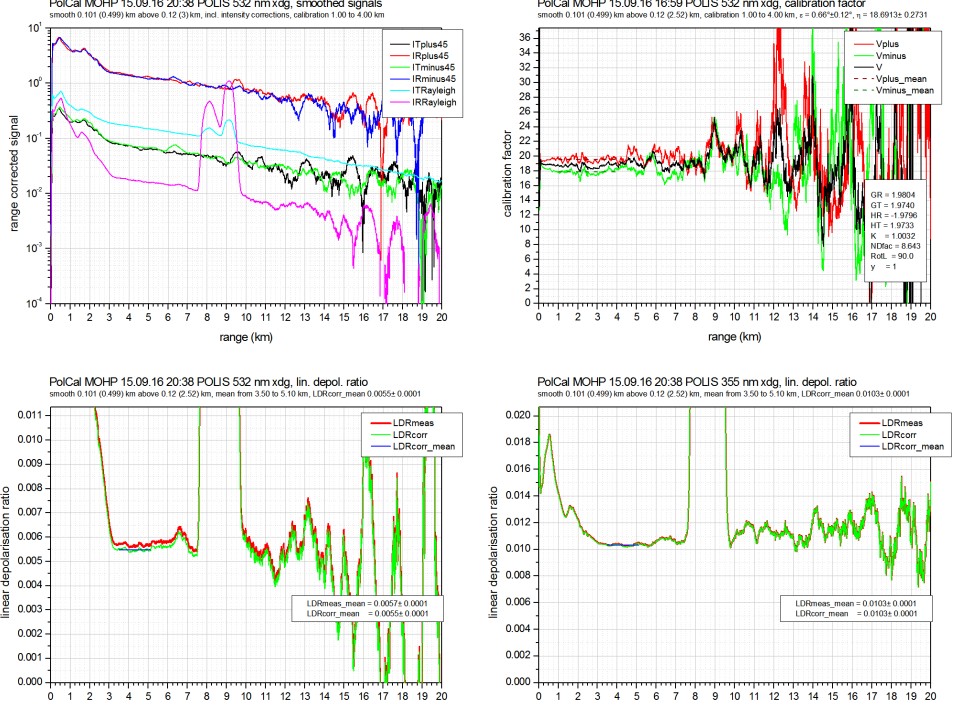

**Figure 34: Top left: Δ90 polarisation calibration and Rayleigh measurements with POLIS-6 at 532 nm. Top right:**
**Retrieved calibration factor. Bottom left: Height profile of the linear depolarisation ratio at 532 nm derived with the**
**calibration factor and the GHK-parameters of the system from the atmospheric Rayleigh measurement (top left) at**
**the same night as the calibration. Bottom right: Same as bottom left but at 355 nm (calibration and GHK-parameters**
**not shown).**
**8    Acknowledgements**
The financial support for EARLINET by the European Commission under grants n. EVR1-CT-1999-40003
(EARLINET, EU - FP5: 2000 – 2003), n. 025991 (EARLINET-ASOS, EU - FP 6: 2006 - 2011), n. 262254
(ACTRIS, EU - FP7: 2011 - 2015) , and n. 654169 (ACTRIS-2, EU - H2020: 2015 - 2019)  is gratefully
acknowledged.



**9  Appendix**
**9.1    Nomenclature of the lidar channel short cuts**
Each lidar channel/signal has a name as **532xtg**, which is composed of the wavelength (in nm) and a three
character (fourth optional) short-cut, which means :
**1st character**
f__ = **f**ar range telescope signal
n__ = **n**ear range telescope signal
g__ = calibrated **g**lue of far range and near range telescope signals
x__ = a single telescope
h__ = **h**igh range signal (single telescope, attenuation adjusted for far range)
l__ = **l**ow range signal (single telescope, attenuation adjusted for near range)
**2nd character**
_t_ = **t**otal signal (no depol. measurement)
_p_ = **p**arallel signal
_c_ = **c**ross signal
_s_ = calibrated **s**um of p and c
_v_ = **v**olume linear depolarization ratio
_a_ = **a**erosol linear depolarization ratio
_e_ = **e**xtinction coefficient
_b_ = **b**ackscatter coefficient
**3rd character**
__a = **a**nalogue signal
__p = **p**hoton counting signal
__g = analogue and photon counting **g**lued signal (e.g. LICEL)
**4th character** (optional)
___l = rotational Raman **l**ower wavelengths
___h = rotational Raman **h**igher wavelengths
___r = **r**otational Raman high and low wavelengths
___c = high spectral resolution Mie signals / **c**entre line



## 9.2 Calculation of the molecular (Rayleigh) signal
The scattering coefficients of air can be calculated in different ways, with various approximations. This variety
of approaches caused some confusion in the past literature. Young (1982) gives a historical overview.
In the electro-dynamical concept the light incident into a volume of air induces oscillating dipole moments in the
air molecules, which re-emit the light independently from each other, i.e. incoherently (besides in exact forward
direction). Miles et al. (2001) give a short but complete overview. van de Hulst (1981) treats the electro-
dynamical aspects of scattering in many details. According to Manneback (1930) (in German) and Young (1982)
the basics of the following relations were already derived by Cabannes and Rocard (1929)  (in French).
Air is a mixture of 78.084 % $N_2$ molecules, 20.946 % $O_2$, 0.935 % Ar, variable small fractions of $CO_2$ and $H_2O$
(Tomasi et al. 2005), and other negligible gases. The main contribution to scattering in air comes from the
diatomic molecules $N_2$ and $O_2$. The following is mainly according to Miles et al. (2001). To model the behaviour
of diatoms in an external radiation field, we use the dipole polarizability tensor α, a tensor of rank 2. Assuming
cylindrical symmetry for diatomic molecules, the invariants of this tensor are the scalars mean polarizability $a$,
and the anisotropy γ. Considering linear polarized laser illumination with vacuum wavelength  λ travelling in the
x-direction of a right-hand Cartesian coordinate system, linear polarized in the y-z plane with an angle β to the z-
axis, the differential scattering cross sections in the direction with angles $\theta_{x,y,z}$ to the x,y,z – axes can be found
for the vertically polarized scattered intensity (index V, polarization in the x-y plane) and for the parallel
polarized intensity (index H, polarization in the x-z plane) by averaging over all molecule orientations, which
yields Eqs. (15)

$$\frac{\partial^{\beta}\sigma_V}{\partial\Omega} = \frac{\pi^2}{\varepsilon_0^2\lambda^4}\left[\left(\frac{3\gamma^2}{45}\right)+\left(\frac{45a^2+\gamma^2}{45}\right)\left(\sin\beta\frac{\cos\theta_y\cos\theta_z}{\sin\theta_z}-\cos\beta\sin\theta_z\right)^2\right]$$
$$\frac{\partial^{\beta}\sigma_H}{\partial\Omega} = \frac{\pi^2}{\varepsilon_0^2\lambda^4}\left[\left(\frac{3\gamma^2}{45}\right)+\left(\frac{45a^2+\gamma^2}{45}\right)\left(\sin^2\beta\right)\left(\frac{\sin^2\theta_z-\cos^2\theta_y}{\sin^2\theta_z}\right)\right]$$
((15)

Integrating over all scattering angles $\theta_{x,y,z}$  we get the total Rayleigh scattering cross section of a  molecule
independent of the input polarization β in Eq. (16)

$$\sigma = \frac{8\pi^3}{3\varepsilon_0^2\lambda^4}\left(\frac{45a^2+10\gamma^2}{45}\right)$$
(16)

which can also be written using the King correction factor $F_k$ describing the anisotropy with Eq. 17.

$$\sigma = \frac{8\pi^3}{3\varepsilon_0^2\lambda^4}a^2F_k$$
(17)

We get the molecular scattering coefficient $\sigma_m$ ,which in absens of absoption is equal to the extinction coefficient,
by multiplying the total Rayleigh scattering cross section with the number density N (molecules / m^3) of air,
which depends on pressure p(z) and  temperature T(z) at height $z$ in the atmosphere

$$\sigma_m = N(z)\sigma = N(z)\frac{8\pi^3}{3\varepsilon_0^2\lambda^4}a^2F_k$$
(18)



The mean polarizability $a$ can be derived accurately from measurement of the refractive index $n$ of air [Ciddor
(2002); Tomasi et al. (2005)] by means of the Lorentz-Lorenz equation (van de Hulst (1981), chap. 4.5)

$$a = \frac{3\varepsilon_0}{N} \frac{n^2 - 1}{n^2 + 2} \qquad (19)$$

Note, as $(n - 1) \ll 1$, there are several confusing approximations used in the literature as

$$\frac{\left(n^2 - 1\right)^2}{\left(n^2 + 2\right)^2} \approx \frac{1}{9}\left(n^2 - 1\right)^2 = \frac{1}{9}\left(n - 1\right)^2 \left(n + 1\right)^2 \approx \frac{4}{9}\left(n - 1\right)^2 \quad \text{and}$$

$$24\frac{\left(n^2 - 1\right)^2}{\left(n^2 + 2\right)^2} \approx \frac{32}{3}\left(n - 1\right)^2 = \frac{4 * 8}{3}\left(n - 1\right)^2 \qquad (20)$$

As the mean polarizability $a$ is a property of an individual molecule, the term on the right hand side does not
depend on the density of the gas, i.e. pressure and temperature, at least for atmospheric conditions. It is usual to
indicate this independence by calculating the mean polarizability at STD air conditions (subscript $_s$ , $=> N_s$ , $n_s$ ).
Considering additionally that the mean polarizability and the anisotropy depend on the wavelength $\lambda$ of the
incident light, we get with Eq. 18 the well known equation for the molecular scattering coefficient $\sigma_m$

$$\sigma_m(z) = N(z)\frac{24\pi^3}{\lambda^4 N_s^2}\frac{\left(n_s^2(\lambda) - 1\right)^2}{\left(n_s^2(\lambda) + 2\right)^2}F_k(\lambda) \qquad (21)$$

with the King correction factor $F_k$

$$F_k(\lambda) = 1 + \frac{2}{9}\left(\frac{\gamma(\lambda)}{a(\lambda)}\right)^2 = 1 + \frac{2}{9}\varepsilon(\lambda) = 1 + \frac{2}{9}R_A(\lambda) \qquad (22)$$

The symbols $\varepsilon$ and $R_A$ for the square of the relative anisotropy are frequently used, e.g. by Kattawar et al. (1981)
and She (2001). The wavelength $\lambda$ represents the energy of the photons and is thus the wavelength in vacuum.
The King factor has been determined very early by measuring the polarization of light scattered perpendicular to
the incident light and using the theoretical relations in Eq. (15) assuming a "mean diatomic air molecule" (see
Young (1982)).
With measurements of the individual air constituents, the King factor can also be determined from a weighted
sum of $F_k$-values according to Bates (1984). Tomasi et al. (2005) compiled the latest fits to measured values of
the wavelength dependent refractive indices and King factors and include the contributions of $CO_2$ and water
vapour in their formulas. These values together with Eqs. (15) and (21) are usually used to calculate the
extinction and backscatter coefficients and the linear depolarization ratio of air for the full Rayleigh scattering,
i.e. the Cabannes line plus the rotational Raman lines, for atmospheric research.
In order to explain the rotational Raman lines, we need a better model. Manneback (1930) derived the basic
formulations already, but unfortunatelly the paper is in German. In the quantum mechanical concept the air
molecules are visualized as rotors, consisting mainly of two ($O_2$, $N_2$, $H_2$) or three ($CO_2$, $H_2O$) atoms. Such rotors
can rotate around different symmetry axes, can vibrate, and the electrons also have momentums and spins.
Incident photons are re-emitted, with or without changing the original state of rotation, vibration, and spins of
the molecule. The quantized energy transfers to the molecules result in discrete wavelength shifts of the emitted
light, i.e. rotational and vibrational Raman scattering (see e.g. Long (2002)). In the quantum mechanical theory



selection rules have been developed based on symmetry considerations of the scattering process and of the
molecular structure, with which the strength of the individual Raman lines can be predicted. But therefore
approximations have to be made. One approximation is the neglect of all transitions but between the states of the
total angular momentum (quantum number J) of the molecules, which gives the pure rotational Raman lines.
Vibrational transitions are not of our concern here  considering the bandwidth of our interference filters of only a
few nm. (Please correct me if formulations are not clear.)
With the latter approximation, the total Rayleigh scattering (superscript [T]) consists of the unshifted and partly
depolarized Cabannes line (superscript [C]), and the wavelength shifted and fully depolarized rotational Raman
lines (superscript [W] used by Young and Kattawar for "wing"; i.e. Stokes and anti-Stokes lines). The Cabannes
line itself contains two components: The weaker component comes from rotational Raman transitions but
without energy transfer to the molecule. Miles et al. (2001) describe this transitions as a reorientation of the
molecules without energy transfer. This part of the Cabannes line is depolarized. The stronger component of the
Cabannes line (Placzek scattering) has no depolarization.
Most papers in literature rely on the formulas given by Kattawar et al. (1981) for the integral strength of these
two parts for high temperatures or $J \Rightarrow \infty$, i.e. the Cabannes line and the sum of the rotational Raman wings (see
e.g. (She 2001) The result is basically the same as from electro-dynamical considerations as in Eq. (15).
Please note, that Eq. (15) for the Cabannes and Rayleigh scattering was derived first from electro-dynamical
considerations, and then also from quantum mechanical theory using approximations and sums over the
Cabannes and wing lines Manneback (1930). For individual rotational lines the full quantum mechanical theory
has to be used (Long 2002) .

### 9.2.1    Backscatter coefficients and linear depolarization ratios

For backscatter ($\theta_{y,z}=90°$) and vertically linear polarized incident light we get from Eq. (15) the differential cross
sections in Eq. (23).

$$\left.\frac{\partial^V \sigma_V}{\partial \Omega}\right|_{180°} = \frac{\pi^2}{\varepsilon_0^2 \lambda^4}\left[\left(\frac{3\gamma^2}{45}\right)+\left(\frac{45a^2+\gamma^2}{45}\right)\right]$$

$$\left.\frac{\partial^V \sigma_H}{\partial \Omega}\right|_{180°} = \frac{\pi^2}{\varepsilon_0^2 \lambda^4}\left(\frac{3\gamma^2}{45}\right)$$
(23)

The term

$$\left(\frac{45a^2+\gamma^2}{45}\right)$$
(24)

comes from the Cabannes line. Multiplying with the number density of air

$$N(z) = \frac{N_A p(z)}{R_a T(z)} = \frac{p(z)}{kT(z)}$$
(25)

(Boltzmann constant k = $1.3806504*10^{-23}$ J/K, Avogadro's number $N_A$ = $6.02214\times10^{23}$ (1/mol), gas constant $R_a$
= 8.314472  (J/K/mol) ) and using Eq. (19) for the mean polarizability $a$, we get for the backscatter coefficients
parallel and perpendicular polarized to the incident polarization, and for the extinction coefficient from Eq. (21)





$$\beta_{m\parallel}^{T}(z,\lambda) = N(z)\frac{9\pi^2}{\lambda^4 N_s^{\,2}}\frac{\left(n_s^{\,2}(\lambda)-1\right)^2}{\left(n_s^{\,2}(\lambda)+2\right)^2}\left(\frac{45+4\varepsilon(\lambda)}{45}\right)$$

$$\beta_{m\perp}^{T}(z,\lambda) = N(z)\frac{9\pi^2}{\lambda^4 N_s^{\,2}}\frac{\left(n_s^{\,2}(\lambda)-1\right)^2}{\left(n_s^{\,2}(\lambda)+2\right)^2}\left(\frac{3\varepsilon(\lambda)}{45}\right) \tag{26}$$

$$\sigma_m(z,\lambda) = N(z)\frac{24\pi^3}{\lambda^4 N_s^{\,2}}\frac{\left(n_s^{\,2}(\lambda)-1\right)^2}{\left(n_s^{\,2}(\lambda)+2\right)^2}\left(\frac{45+10\varepsilon(\lambda)}{45}\right)$$

with

$$\varepsilon(\lambda) = R_A(\lambda) = \frac{\gamma^2(\lambda)}{a^2(\lambda)} \tag{27}$$

Please note that the refractive index $n_s$ at standard air conditions is only used to calculate the mean polarizability of an air molecule with Eq. (18); for nothing else. Comparing with Kattawar et al. (1981) we see that the Cabannes and wing intensities are proportional to

$$\beta_{m\parallel}^{C}(z,\lambda) \propto \left(\frac{45+\varepsilon(\lambda)}{45}\right) = \frac{180+4\varepsilon(\lambda)}{180}$$

$$\beta_{m\parallel}^{W}(z,\lambda) \propto \left(\frac{3\varepsilon(\lambda)}{45}\right) = \frac{12\varepsilon(\lambda)}{180}$$

$$\beta_{m\perp}^{C}(z,\lambda) \propto \left(\frac{\tfrac{3}{4}\varepsilon(\lambda)}{45}\right) = \frac{3\varepsilon(\lambda)}{180} \tag{28}$$

$$\beta_{m\perp}^{W}(z,\lambda) \propto \left(\frac{\tfrac{9}{4}\varepsilon(\lambda)}{45}\right) = \frac{9\varepsilon(\lambda)}{180}$$

from which we get backscatter coefficient components for the Cabannes and total Rayleigh scattering for linear polarized incident light

$$\beta_m^{C}(z,\lambda) = N(z)\frac{9\pi^2}{\lambda^4 N_s^{\,2}}\frac{\left(n_s^{\,2}(\lambda)-1\right)^2}{\left(n_s^{\,2}(\lambda)+2\right)^2}\left[\frac{180+7\varepsilon(\lambda)}{180}\right]$$

$$\beta_m^{W}(z,\lambda) = N(z)\frac{9\pi^2}{\lambda^4 N_s^{\,2}}\frac{\left(n_s^{\,2}(\lambda)-1\right)^2}{\left(n_s^{\,2}(\lambda)+2\right)^2}\left(\frac{21\varepsilon(\lambda)}{180}\right)$$

$$\beta_m^{T}(z,\lambda) = N(z)\frac{9\pi^2}{\lambda^4 N_s^{\,2}}\frac{\left(n_s^{\,2}(\lambda)-1\right)^2}{\left(n_s^{\,2}(\lambda)+2\right)^2}\left[\frac{180+28\varepsilon(\lambda)}{180}\right] \tag{29}$$

$$\sigma_m(z,\lambda) = N(z)\frac{24\pi^3}{\lambda^4 N_s^{\,2}}\frac{\left(n_s^{\,2}(\lambda)-1\right)^2}{\left(n_s^{\,2}(\lambda)+2\right)^2}\left[\frac{180+40\varepsilon(\lambda)}{180}\right]$$

$$F_k(\lambda) = \frac{180+40\varepsilon(\lambda)}{180}$$



We get the Cabannes and total Rayleigh linear depolarization ratios from

$$\delta_m^T(\lambda) = \frac{\beta_{m\perp}^T(\lambda)}{\beta_{m\parallel}^T(\lambda)} = \frac{12\varepsilon(\lambda)}{180+16\varepsilon(\lambda)} = \frac{3F_k(\lambda)-3}{4F_k(\lambda)+6} \Rightarrow F_k(\lambda) = \frac{3+6\delta_m^T(\lambda)}{3-4\delta_m^T(\lambda)}$$
(30)

and

$$\delta_m^C(\lambda) = \frac{\beta_{m\perp}^C(\lambda)}{\beta_{m\parallel}^C(\lambda)} = \frac{3\varepsilon(\lambda)}{180+4\varepsilon(\lambda)} = \frac{3F_k(\lambda)-3}{4F_k(\lambda)+36}$$
(31)

The following is equivalent to Hostetler and Coauthors (2006) in order to use this paper as reference. We write
using Eq. (18) and Eq. (25) the total Rayleigh scattering cross section

$$\sigma_m(z,\lambda) = C_s(\lambda)\frac{p(z)}{T(z)}$$
(32)

and the backscatter coefficients for the Cabannes line and total Rayleigh scattering

$$\beta_m^{C,T}(z,\lambda) = \frac{\sigma_m(z,\lambda)}{S_m^{C,T}(\lambda)} = \frac{\sigma_m(z,\lambda)}{8\pi\big/3\, k_{bw}^{C,T}(\lambda)} = B_s^{C,T}(\lambda)\frac{p(z)}{T(z)}$$
(33)

with the lidar ratio S

$$S_m^{C,T}(\lambda) = \frac{8\pi}{3}k_{bw}^{C,T}(\lambda)$$
(34)

and get the conversion factors

$$k_{bw}^T(\lambda) = \frac{3}{8\pi}\frac{\sigma_m(\lambda)}{\beta_m^T(\lambda)} = \frac{180+40\varepsilon(\lambda)}{180+28\varepsilon(\lambda)} = \frac{10F_k(\lambda)}{7F_k(\lambda)+3} = \frac{1+2\delta_m^T(\lambda)}{1+\delta_m^T(\lambda)}$$
(35)

$$k_{bw}^C(\lambda) = \frac{3}{8\pi}\frac{\sigma_m(\lambda)}{\beta_m^C(\lambda)} = \frac{180+40\varepsilon(\lambda)}{180+7\varepsilon(\lambda)} = \frac{40F_k(\lambda)}{7F_k(\lambda)+33} = \frac{1+2\delta_m^T(\lambda)}{1-\dfrac{3}{4}\delta_m^T(\lambda)}$$
(36)

Note that Eq. (4.15) in Hostetler and Coauthors (2006), which reads

$$k_{bw}^C(\lambda) = \frac{3}{8\pi}\frac{\sigma_m(\lambda)}{\beta_m^C(\lambda)} = \frac{180+40\varepsilon(\lambda)}{180+10\varepsilon(\lambda)} = \frac{4F_k(\lambda)}{F_k(\lambda)+3} = \frac{1+2\delta_m^T(\lambda)}{1-\dfrac{1}{6}\delta_m^T(\lambda)}$$
(37)

is wrong. However, in the actual CALIPSO data analysis the correct conversion factors are used according to
Powell et al. (2009)





| wavelength (air/vacuum) [nm] | $(n_s - 1)$ [*1e-8] STD air | King factor $F_k$ STD air | $C_s$ (32)(29)(25) [K/hPa/m] | $B_s^T$ (33) [K/hPa/(m*sr)] | $B_s^C$ (33) [K/hPa/(m*sr)] | $k_{bw}^T$ (35) | $k_{bw}^C$ (36) | $\sigma_m$ (32) [1/m] STD air | $\beta_m^T$ (33) [1/(m*sr)] STD air | $\beta_m^C$ (33) [1/(m*sr)] STD air | $\delta_m^T$ (30) [*1e-2] STD air | $\delta_m^C$ (31) [*1e-2] STD air |
|---|---|---|---|---|---|---|---|---|---|---|---|---|
| 308 / 308.089 | 29046.6 | 1.05574 | 3.6506e-5 | 4.2886E-6 | 4.1678E-6 | 1.01610 | 1.04554 | 1.2837E-4 | 1.5080E-5 | 1.4656E-5 | 0.01636 | 0.004158 |
| 351 / 351.100 | 28602.7 | 1.05307 | 2.0934e-5 | 2.4610E-6 | 2.3949E-6 | 1.01535 | 1.04338 | 7.3611E-5 | 8.6539E-6 | 8.4214E-6 | 0.01559 | 0.003959 |
| **354.717 / 354.818** | 28572.4 | 1.05290 | 2.0024E-5 | 2.3542E-6 | 2.2912E-6 | 1.01530 | 1.04324 | 7.0414E-5 | 8.2783E-6 | 8.0566E-6 | 0.01554 | 0.003946 |
| 355 / 355.101 | 28570.2 | 1.05288 | 1.9957E-5 | 2.3463E-6 | 2.2835E-6 | 1.01530 | 1.04323 | 7.0177E-5 | 8.2506E-6 | 8.0393E-6 | 0.01554 | 0.003946 |
| 386.890 / 387.000 | 28350.2 | 1.05166 | 1.3942e-5 | | | | | 4.8925E-5 | | | | |
| 400 / 400.113 | 28275.2 | 1.05125 | 1.2109E-5 | 1.4242E-6 | 1.3872E-6 | 1.01484 | 1.04191 | 4.2579E-5 | 5.00810E-6 | 4.8780E-6 | 0.01507 | 0.003825 |
| 407.558 / 407.673 | 28235.1 | 1.05105 | 1.1202e-5 | | | | | 3.9389E-5 | | | | |
| 510.6 / 510.742 | 27869.4 | 1.04922 | 4.4221E-6 | 5.2042E-7 | 5.0742E-7 | 1.01427 | 1.04026 | 1.5550E-5 | 1.8300E-6 | 1.7843E-6 | 0.01448 | 0.003673 |
| 532 / 532.148 | 27819.9 | 1.04899 | 3.7382E-6 | 4.3997E-7 | 4.2903E-7 | 1.01421 | 1.04007 | 1.3145E-5 | 1.5471E-6 | 1.5086E-6 | 0.01441 | 0.003656 |
| **532.075 / 532.223** | 27819.4 | 1.04899 | 3.7361E-6 | 4.3971E-7 | 4.2878E-7 | 1.01421 | 1.04007 | 1.3138E-5 | 1.5462E-6 | 1.5078E-6 | 0.01441 | 0.003656 |
| 607.435 / 607.603 | 27686.3 | 1.04839 | 2.1772e-6 | | | | | 7.6559E-6 | | | | |
| 710 / 710.196 | 27570.4 | 1.04790 | 1.1561E-6 | 1.3611E-7 | 1.3280E-7 | 1.01390 | 1.03919 | 4.0655E-6 | 4.7863E-7 | 4.66698E-7 | 0.01410 | 0.003575 |
| 800 / 800.220 | 27503.8 | 1.04763 | 7.1364E-7 | 8.4022E-8 | 8.1989E-8 | 1.01383 | 1.03897 | 2.5094E-6 | 2.9546E-7 | 2.8831E-7 | 0.01402 | 0.003555 |
| 1064 / 1064.292 | 27397.5 | 1.04721 | 2.2622E-7 | 2.6638E-8 | 2.5999E-8 | 1.01371 | 1.03863 | 7.95949E-7 | 9.3670E-8 | 9.1423E-8 | 0.01390 | 0.003524 |
| **1064.150 / 1064.442** | 27397.4 | 1.04721 | 2.2609E-7 | 2.6623E-8 | 2.5984E-8 | 1.01371 | 1.03863 | 7.9504E-7 | 9.3617E-8 | 9.1371E-8 | 0.01390 | 0.003524 |

**Table 1: Refractive index ($n_S$), King factor ($F_k$), extinction coefficients ($\sigma_m$), Cabannes ($\beta_m^C$) and total**
**Rayleigh ($\beta_m^T$) backscatter coefficients, proportionality factors (see text above), and Cabannes ($\delta_m^C$) and**
**total Rayleigh ($\delta_m^T$) linear depolarisation ratios caclulated with the equations in row two, for STD air**
**conditions where mentioned (STD air: $p_s$ = 1013.25 hPa, $T_s$ = 288.15 K). The refractive indices and the**
**King factors are calculated according to Tomasi et al. (2005) and Ciddor (2002) with 385 ppmv $CO_2$ and**
**0% RH. Please note that the values in the table of the Tomasi paper were caclulated for slightly different**
**conditions. NdYAG elastic and Raman wavelenghts (underlined) are for vacuum, calculated from the**
**fundamental air wavelength 1064.15 nm (1064.442 nm in vacuum) at 300 K rod temperature according**
**to Kaminskii (1990). In order to enable the comparison of the accuracy of the calculations by the readers,**
**more decimal digits are shown than certified by the accuracy of the model and the assumtions.**



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
