# Peer review of "EARLINET lidar quality assurance tools"

_Atmospheric Measurement Techniques, 2017_

## Referee Comment (RC1) · Anonymous Referee #3 · 30 Jan 2018

The paper entitled "EARLINET lidar quality assurance tools" by Volker Freudenthaler, Holger Linné, Anatoli Chaikovski, Dieter Rabus, and Silke Groß describes several tools and tests for the monitoring of lidar hardware setup and estimation of uncertainties of derived products.

———————————— general comments ————————————

The paper is of scientific significance, it provides a variety of valuable tools and methods for uncertainty estmimation for lidar scientists. The scientific approaches and applied methods are valid. Nevertheless, the presentation of the methods and results can be improved.

The first sentence of the introduction should be deleted or re-phrased. It is uncommon to say that the introduction to the actual paper is provided in chapter 3.1 of another paper. The authors could at least give a short summary of this section.

[Figure]

introduction: The text should be structured in several paragraphs

p1,l36 - p2, l14: The introduction should provide an outlook to the content of the following sections. There are too many technical details and results provided in this part of the manuscript.

All figures are too small. Legends are really hard to read.

The manuscript misses a summary and/or conclusion section.

———————————————— Specific comments ————————————————

introduction: p1, l18: why are the instruments not homogeneous?

p1, l24-25: The phrase "which was .. ozone research" is not needed here.

p2, l1: a too high signal-to-noise ratio - in near range?

fig1, caption: .. red curves are range corrections -> are range-corrected signals

section 2.1: Please consider to shorten this section. It should be sufficient to provide equations (1),(2), and (4) together with a reference to the extinction retrieval which is available, e.g., in lidar textbooks.

eq (1): the symbols $\beta$ and $C$ are not explained in the text.

p3, l19: provide few more words how the differentiation is related to the absolute error.

section 2.2: A list of the several discussed methods would be helpful as introduction.

eq(6)/fig2: how is the delta alpha of figure 2 related to equation(6)?Equation(6) cannot give negative values.

p5, l5-6: what does "decrease the signal height .. detectors" mean? Shall the aperture suppress signal saturation?

p5, l12-13: Which accuracy of the speed of light in the fibre is needed to measure the trigger delay with an accuracy of 7.5m?

p6, l8: "The main peaks are distributed between two range bins." -> Figure 7 shows that the peaks of the photon counting signal are distributed over 3-4 bins.

p6, l8-9: "The statistical properties... better than a range bin" -> Provide details / equations how this uncertainty can be derived or skip the sentence.

p6, l22: "The small correlation peak" -> which one? The peak at 0 is larger than the one at -11?

p7, l5: It could be emphasized that you now start the discussion of another (traditional, widely used) method. Otherwise the reader may not realize that figure 7 is about lab measurements and figure 10 about atmospheric measurements.

Figure 9 could be removed because its information is also provided in figure 10.

section 3: This section needs re-organization. It should start with a general description of the method (equations 7-9), followed by the discussion of measurement examples and a conclusion. This section mixes the discussion of the accuracy of the Rayleigh fit (resulting in calibration uncertainties) with the use of the Rayleigh signal as a tool for the quality check of signals (analog signal distortions). These two aspects of the Rayleigh fit should be discussed separately.

p9, l4-6: This sentence is difficult to read.

eq(8): $\beta_p$ should be $\beta_m$

eq(10): $r^2 P(r, r_0)$ should be $r^2 P(r)$

eq(10): it should be mentioned that the term for particle transmission is completely neglected in equation (8) and the following.

eq(11) is wrong. $\beta_m^{att}$ and $\beta_m$ are not equal. They are different by the molecular transmission term.

eq (11): The interesting question is how the uncertainties of the normalization

(Rayleigh fit) propagates to the uncertainty of the retrieved backscatter profiles? Which uncertainties of the backscatter profiles are to be expected if the normalization range is not completely free of particles? It is not necessary to introduce new simulations here, but corresponding values from literature could be helpful to understand the importance of a good Rayleigh fit.

p9, l16-20: These approximations of the backscatter ratio and backscatter coefficient should be skipped because they are not used in the following parts of the manuscript.

fig11: You should show only one Rayleigh fit. It is confusing to show different options and not to discuss which fit range is optimum.

p8, l12: What is the glued signal? Skip this curve. Gluing is not the scope of this manuscript.

It should mentioned that the analog signals discussed in fig 11 and fig 12 are different. The one in fig 11 (Licel) is not optimized for the far-range in contrast to one in fig 12. It is worth to mention as a conclusion that those Licel analog signals should never be used without the corresponding photon-counting signal.

Why is the uncertainty of the analog signal in Fig 11 calculated with respect to the photon counting signal and in fig 12 with respect to the calculated Rayleigh signal? Which method is recommended?

fig 13: could be skipped because there is no additional information to figure 12.

fig16: The discussion of Licel curves is quite short. E.g., why do cyan lines have a stronger noise than black and magenta lines?

figures 16 and 17: The legends are too small and the numbers are difficult to understand. A legend in table form beside the plot could significantly improve readability.

fig24: would be easier to read if the nomenclature is plotted left. What is the viewing direction - towards the atmosphere or into the lidar?

fig24,caption: skip "Using the four .. octant test". These terms are explained in the text; "the pictures above.." -> pictures left

p17, l8: insert something like "The measurement sequence started with the north sector followed by ..."

p17, l13: "due o signal noise, except for .." -> due to signal nose and atmospheric ...

p18, l1 : Is there a threshold at which deviations the overlap is considered as full? It is really hard to see the curves below 100m. A break of the axis at 1-4km would enlarge the overlap range.

fig 26: The optical schematics are too small! Are they needed to understand the simulations? what are the green and blue areas in the left plot?

fig 27: legends are not readable.

figs 28-30: skip titles. If necessary explain in the caption. What are symbols and lines?

p19: Are there differences between the results of the paraxial and the ZEMAX simulation? If yes, discuss the pros and cons of the two simulation tools. If no, skip one method.

figs 31-32: skip titles. If necessary explain in the caption.

annex: The variable $\beta$ is used as an angle in eq (15) and as backscatter coefficient.

———————————— technical corrections ————————————

p5, l11: "With this" -> with this setup

p6, l15: Figures 7 -> Figure 7

p8, l2: "calculated signals from air density" -> calculated pure molecular signals

fig11: skip the title of the plot

fig12: skip titles of the plots

p8, l21: without aerosols -> without aerosol particles

p8, l25: What does "too strong" mean? E.g. larger than 5

p11, l2-3: "..clearly visible in section 3 about .. above." -> "..clearly visible in section3."

p16, l2: "deviations of" -> deviations between
* * *

---

## Referee Comment (RC2) · Anonymous Referee #2 · 28 Mar 2018

The manuscript "EARLINET lidar quality assurance tools" by V. Freudenthaler et al. summarizes the tools that have been developed approximately over the last decade to harmonize EARLINET lidar systems and their data products. The paper mainly focusses on checks according to hardware issues (and not on the quality of the data analysis algorithms). That fact should probably already be addressed in the title.

I believe the manuscript is an important contribution to the lidar community and hence it should be published.

However, up to know this manuscript is sometimes a bit confusing. I would suggest several points to consider before publication:

-The introduction reads a bit unmotivated. Maybe it does not need to be long but the general idea to introduce into the topic, to cite work that has been done before, to show the significance and relevance, and to outline the manuscript should be taken into

account. It should also be a bit more structured, e.g., suddenly at p1l36 the common issues like trigger delay are mentioned, these technical details should be introduced and explained, at least in a way, that they will be treated in this manuscript further below.

-Maybe there should also be a short overview chapter on the known lidar hardware issues (if it cannot be given in the introduction) and on the quality tests. Right now, the start with section "2 Trigger delay" is abrupt.

-All figures need attendance. The labeling and often also the line width is too small by at least a factor of 3 to 5. Some figures contain cryptic (at least for the common reader) abbreviations. Also, the quality is quite poor on some graphics. It should be checked if vector graphics could be used.

-Some statements maybe lack citations, I will try to mention a few of those below.

-The manuscript text should be true BLACK, on my printer it was just an RBG mixture and thus all letters had a colorful shade around.

-There are quite a few sentences which need some language rephrasing. Please consider rechecking the manuscript carefully with respect to that. Specific comments:

Sect. 1:

-P1L30. I think the sentence should be rephrased, also "wherefore"(several times used in the manuscript) seems to be an antique word nowadays, e.g.: Nonetheless, a direct lidar intercomparison with a reference system is considered to be the ultimate test because it can often reveal problems that have been missed even after the tests described in this manuscript have been performed.

-P2L3. Photon-counting saturation is mentioned here as a possible problem but never mentioned in the manuscript. Please try to make clear which problems will be addressed in the paper and which not. Also, include a citation on the photon-counting saturation. And maybe also on the other effects. Sect. 2:

-P2L17. "Large errors up to 1 km" What means large and why up to 1 km. Maybe the conditions should be put first. "Assuming the zero bin error to be of 15 m errors on the order of ... can be as large as ..% in the lowest 1 km. Furthermore, the error of the extinction can be as large as 1 km-1 at a height of 100 m."

-Fig1-4. In general, I think the whole story could be better presented in a 4-panel graph from one single measurement case. As of now, Fig. 1 is just some independent, arbitrary data case. My suggestion: Left top and bottom: Signal and RC-Signals with different trigger delays (like Fig 1 but with data from Fig. 3). Right top: Absolute Extinction error (Fig. 2), Right bottom: Extinction coefficients (Fig. 4).

-For Fig 1 it is not mentioned, when and where the measurement was taken, therefore I suggest to skip this case as mentioned above.

-Fig1. I would suggest marking the "0" on the x scale, maybe with a vertical bar. The legend m532xcg can be removed. It is just confusing.

-Eq.1,3,4,5 {[()]} the bracket order not kept

-Eq 2, is there a citation?

-Eq 3 and 4 seem to have different fonts.

-Fig2, caption: @355/387, please write properly, e.g., ...from (N_2 vibration-rotation) Raman measurements at 355 nm.

-P3L3. beta_IR should be beta(l,r) in the text, also later the notation should be consistent

-P3L16. A variation/error ... please decide for one or replace "/" with "or an"

-P4L5. Please give some reference to the used lidar systems if possible

-P5L13. Can you mention the exact fiber type and the refractive index? How exact has n to be known? Couldn't the refractive index (speed of light within the fiber) also be

derived by combining two exactly same length fibers and measure the time delay? Or with a photodiode/oscilloscope measurement? Would that maybe more precise?

-P6L2: What could you do, if the range resolution is 30 or even 60m? Is it possible to adjust the fiber length until the peak jumps from one bin to the next?

-Fig 5 and Fig 6 could be combined, as they are directly linked.

-Fig 7 has LSB written on the left scale, but the caption says it is on the right scale P6L22, can you a bit more specify the terms signal noise and background noise here? Does it mean the signal needs to be larger than the background? And how can the peak at 0 in C be explained?

-Fig 8, as the range bins start at 0, is plot A already corrected for the different trigger delays? Please mention in the text.

-I would skip Fig 9 because it is not clear from the text why it is presented. And Fig 7 and 10 can be combined in a two-panel graph. Again, they present the same topic, where first the focus is on shot-to-shot trigger jitter and the second is on inter-channel jitter or delay (cross-channel vs parallel and 607) and on the error made when using the scattered light from the laboratory. Probably even Figure 10 should be presented first (top/left) and Fig 7 should be second (bottom/right) because the near-range peak is a well-known feature to a lidar operator. Can you comment on the size of your lab and if it indeed can explain the 37 m (10 bins) difference between those peaks?

Sect. 3:

-In general, I would place the Rayleigh-fit section after the telecover section. This is an atmospheric test with real measurements, whereas the other tests are for signal conditioning or of laboratory character.

-P8L2, Please explain why the Rayleigh fit is the only "absolute" calibration? Either give some citations or rephrase.

-P8L4, what is a high dynamic range, and doesn't that depend on the wavelength and aerosol conditions?

-Fig 11, the 7 different lines and colors are not visible

-P8L9, dead-time corrected. How was this done?

-P8L9-12, the log plotting info could go into the figure caption, not to distract from the topic. Also, this part is more about gluing than about the Rayleigh fit. Actually, after Fig 11 the text should continue with P8L20.

-P8L10, the term 19 Least Significant Bits should be explained? Isn't the LSB usually the one or the two which are considered to be the noise of the signal?

-Fig 12 or corresponding text, citation for MULIS missing. Why is the Rayleigh fit performed on an A/D signal, when later in the manuscript it is shown that A/D signals can be highly distorted in the weak-signal regime?

-Eq. 7, please mention that this is the elastic lidar equation in contrast to Eq. 1.

-Fig 13. There is not mentioned, why Fig 13 is in the manuscript? What can be seen here that presents some new facts? Could be skipped or combined with Fig 12. However, to demonstrate the effect of analog and p.c. in the far range one graph is enough.

-P10L11-14. Again, many error sources are mentioned but not discussed or analyzed or referenced.

Sect. 4:

-P10L16. This subsection consists of just one sentence. Please elaborate/cite more about sources of analog distortions and introduce in Sect. 4, e.g., that there are electronic and optical test methods. In general, this entire chapter needs some more attention.

-Fig 15 is not mentioned in the text.

-Fig 16+17: I would suggest combining them (one graph, two panels). Also for Fig 16, not all transients have to be shown. After all, this paper shall present the test ideas, not the results of one individual system. Therefore, it would be enough to show one curve each for LICEL and the two SPECTRUM cards. The inter-channel variations are not discussed in the text anyways.

-P13L18. What is the analog output of an A/D converter? Do these modules deliver an analog signal as well or do you mean the digitized output? Please also specify the manufacturer of these modules and the later mentioned detectors.

-Subsect 4.3: This subsection needs much more explanation. What is measured, why are there two pulses (8:1), what does the background LED do? What are F-in levels? What is Vout/Vin, when it is an optical test? Also, these graphs differ in style from the rest of the manuscript.

-Can there anything be said about the EM-distortions in those modules? It is stated that everything (detector, amplifier, A/D converter) is in one box to avoid such distortions. But can this fact be shown as well?

-The end of section 4 should introduce/motivate into the next section of dark measurements.

Sect. 5:

-P15L3-4: Please split such long sentences, because these are two independent effects.

Sect. 6:

-P16L2-4: This is just one sentence in the paragraph

-Fig 22. Please mention that the simulations are for a biaxial lidar setup. What are the top and bottom parts of the telescope? Do you refer here to North and South section

as stated later?

-P16L23. "In a first attempt" ...to do what? Better formulate in a way that in the following paragraph a method is shown to test the effects mentioned above.

-Fig 24, right. Somehow mention the viewing direction. Is the laser coming towards the reader or is it a view from behind the lidar?

-Fig 25. This graph is unnecessarily crowded. I think it is not needed to show the raw signals. In my opinion, the normalized telecover signals and deviations of one channel would be enough. It would be even more informative if next to the good example of POLIS-6 a failed-test example would be shown and the reasons would be explained later in the text.

-P18L2. What means "well-designed eyepieces"? Please give more insight or write well-designed receiver optic.

-P18L6. Since there are two different blue colors in the figure the sentence should also mention "(right)" somewhere.

-Fig 26. left: The blue part should probably also be green. It is not clear why the bar to the left is blueish and the bar to the right is greenish. It is also unclear, why all the laser and receiver optics are named, and what those notations mean. This lower part could be easily skipped, to explain the telescopes overlap effect.

-Fig 27 is very hard to read and understand. First of all, this figure shall explain the effect of telescope defocus. But the section is about the Telecover Test. So it could be argued, why is this off-topic presented here? This must be explained in the text. Secondly, it is not explained, what the new orange and purple colors in the paraxial simulations mean. Where do they come from suddenly? If this graph is really needed, maybe it can be condensed into 3 field-stop positions and just one range from 0-2 km?

-P19L8-12. "Loss of full overlap of the S-signal in the far range." Where can this be seen in Fig 28? This is not obvious.

-Fig 28. Can there be more rays used for the simulation to decrease the noise? Can you describe in the caption what is seen right and left? Why is there mentioned that polarization was considered? How is this relevant to this figure and the statements made?

-P20L4. Please write firstly what is seen in Fig 29, before mentioning disadvantages of the telecover test. It reads so negatively. And what exactly means S-N/S tilt?

-P20L14-18. This sentence is very hard to understand. Can you please explain the arguments a bit more in detail for a reader who is not performing optical simulations on a daily basis? Where is the collimator lens?

-Fig 29. What means coating IFF2C5555...? What is appendix "r"? If there are so many curves in a graph, they need some kind of annotation.

-P21L6FF. It is really hard to follow in this part of the text. Why suddenly the formula Eq. 14 of a transmission shift vs. angle of an interference filter is presented?

-Can Fig 31 and 32 be combined, as they deal with the topic of PMT inhomogeneity?

-Fig 32. What means eyepiece? Can you reference or explain in the text?

-General remarks: Can the telecover tests be used to estimate systematic errors? Let's assume you use different sectors of the telescope to derive backscatter, extinction, depolarization. Can the variability between these products be linked to the systematic error of those products? If such conclusions could be drawn from this test, this would be a real advancement. Like presented now, the telecover test can only discriminate into good alignment or problematic alignment. But what are the consequences? Stop measurements at all until all systems are perfect or adjust error bars?

Sect. 7:

-Fig 34. Vplus_mean and Vminus_mean are not visible.

After Sect 7. I agree with Reviewer #3 that at least a conclusion/summary section is

missing. This would be the ideal place to summarize the recommendations and stan-dardizations requirements given by EARLINET and future ACTRIS for the necessary check-up routines to comply with the quality assurance.

Appendix 9.1: This part could be skipped if the signals in the few graphs would be annotated in a more readable form. If it is kept, it should be mentioned in the text, why this naming convention is used throughout the manuscript.

9.2: It is important that within EARLINET the Rayleigh scattering algorithms are har-monized. However, does this information belong to the context of this -mainly hardware related- manuscript? Then it could also be argued, why different algorithmic solution approaches of the lidar equations are not summarized here, too? I would suggest to skip this part and publish it elsewhere. If not, then again, it should be mentioned in the text, why this Appendix is needed here.

Literature: There are quite some typos, I guess this will be fixed later.

---

## Referee Comment (RC3) · Anonymous Referee #1 · 6 Apr 2018

The paper "EARLINET lidar quality assurance tools" describes check-up tools, tests and methods for improving of the lidar systems and data products.

General Comments The paper is of scientific significance within the scope of this journal. The manuscript represents important new tools and methods for tests of lidar systems and improvement of data quality. The scientific approaches and applied methods are valid. The results are discussed in an appropriate way. Presentation quality can be improved.

The "Abstract" should include some of the key results. I suggest re-writing of "Introduction" section. It should be organized in several paragraphs. It is unusual the introduction of a present paper to be a section from another paper published 4 year ago. The values of the numbers should be limited to the significant digits only. All the figures can be enlarged. Axis labels and legends are hard to read. Conclusions/Summary section is

missing.

Specific comments p2, Section 2, paragraph 1, Authors should provide more information about origin of the trigger delay between the actual laser pulse and zero-bin. The delay can originate from trigger source and/or design of the transient recorder. Trigger source usually is one of: laser flash lamp out pulse, laser Q-switch out pulse or a photodiode picking up a fraction of the outgoing beam. Usually triggering by a photodiode ensures minimum trigger delay and fluctuations.

The bin shift between photon counting and analog signals is an expected result caused by specific of Analog to Digital Convertors. For example, Licel provide information about factors for bin shifting of their transient recorders as well as software for correction of delay/shift between photon counting and analog signals. The information can be found on the web in section "9.5.6 Tutorial" in "Lidar transient recorder" manual (http://licel.com/manuals/ethernet_pmt_tr.pdf). I suggest authors to comment on the information provided by manufacturer.

p2/3, Section 2.1 The equations and symbols into the text have inaccuracies that I assume are typos. Please see details below.

p3,L1 (eq.1) in the equation "P(ïĄňR,r)" is used for received power. Term is "PïĄňR" in the text. Symbols in the equation are different from symbols in the text also for Raman backscatter coefficient, extinction coefficients of air molecules and aerosol particles.

p3,L7 (eq.2). The power "4.085" should be negative "-4.085"

p3,L8 The value of parameter fp is 1.09 according to the eq.2. In the text value is reciprocal. "ïĄňLaser" should be "ïĄň0", "ïĄňRaman" should be "ïĄňR"

p3,L9 (eq.3) The term "[1+fm]" should be "[1+1/fp]". The term "[1+fp]" should be "[1+1/fm]".

p3,L15 (eq.4) and L18 (eq.5) The term "(1+fp)" should be "(1+1/fp)". The term "(1+fm)" should be "(1+1/fm)".

p3,L19-23 (eq.6)and p4,Figure 2. The common definition of absolute error is the difference between the measured or inferred value of a quantity and its actual value. Please provide more information about the differentiation approach. The Eq.6 has unit 1/mˆ2 or 1/kmˆ2, while the unit on delta alfa in the figure 2 is 1/km.

p4, Figure 2 should be discussed in the text.

p4, L2, Sentence starting with "Note: as the error. . ." is not clear. Please rephrase it.

p5, section 2.2 "How to measure the trigger delays". It will be good if authors include into this section some results from a triggering by fast photodiode placed near to the laser output. All the results in the section are based on LICEL transient recorders only. Please comment on applicability of the method for other type ADC/photon counting equipment.

p5,L6, ". . .to decrease the signal length . . ." should be ". . .to decrease the signal intensity . . ."

p6,L2, The range can be measured in steps with much better time/range resolution by fast oscilloscope. The signal from PMT can be electronically splited and measured simultaneously by LICEL transient recorder and an oscilloscope. For example, a 200MHz, 5GS/s oscilloscope allows 1ns or 15cm lidar range resolution.

p6,L4, ". . .several electronic delays. . .". Can you please give a little more info?

p6,L8, ". . . are distributed between two rangebins." A better time resolution is required to conclude about main peak distribution.

p6,L11, Figure 7 caption. "Trigger delay / zero-bin. . ." is better to be "Trigger delay (zero-bin). . .".

p6,L15-16, Please include that the delay is an expected result according to the manufacturer. One can conclude that delay is fixed for individual module base on fig.7 and fig.8

p6,L22 "The small correlation peak in plot C. . ." should be "The correlation peak at -11 lag is smaller than peak at 0 lag in plot C because of the noise. . . . . ."

p.8, Section 3 "Rayleigh fit". Authors should include a separate paragraph/subsection regarding to calculation of the Rayleigh signal from a radiosonde data. The resolution of radiosonde data is much smaller than lidar range resolutions which require extrapolation and/or fit of sonde data. It will be good to include more information about this process together with typical errors and uncertainties.

p8,L9 and Figure 11, Please provide more details about photon counting unit "counts/rangebin". A reader can confuse about this units. SI unit for photon flux is "count per second" and usually Mc/s (million counts per second) is used. The Licel acquisition software use so called "MHz" that is count normalized on rangebin (25ns @40MHz digitizer) and number of laser shots. I think only glued signal can be shown on figure 11. The combination of the analog and photon counting is well described in Licel's user manual and a reference about that should be sufficient.

p8,L10-11, It is not clear why LSB units are used in additional y-scale. The analog signal is measured in "mV". ". . ..level of the analog signal." In fact on the figure 11 is present the analog signal but scaled to the photon counting signal, not original one. This is so called by Licel "scaled analog signal". The sentence "The analog signal is at the lowest bit. . ..". If the lowest bit limit is at 2.8km then how is measured the analog signal from 2.8km to 14km?

p8,L12, Please provide more information about what is the "glued signal " and give a reference about glue procedure.

p8,L14-L15,and L20-L22 "The uncertainty of the fit. . ..". This discussion should be after the details of fitting procedure (Eq.7-Eq13).

p8,L22, ". . . fits the lidar signal sufficiently good." Authors should provide a numerical criterion of "sufficiently good".

p8,L22, and Figure 12. Why only analog signal is shown? I suggest using of glued signal. This is the real advantage of simultaneously measurements of both analog and photon counting signals. If authors would like to discuss on limitations on analog signal only as well as on limitations of PC signal only that should be done in a separate section. p9,L4-L19, This is in fact detailed description on fitting procedure and should be moved to the top of the Section 3. p9,L5, "..attenuated molecular backscatter coefficient ". It is a profile according to definition by Eq.8. I think "range corrected Rayleigh signal calculated from radiosonde or standard atmosphere data" is a better name for this term. Also, the term "..attenuated molecular backscatter " is common used in High Spectral Resolution Lidar technique for profiles attenuated by BOTH molecules and particles. p9,L8, (eq.8), "$\beta p(r)$" should be "$\beta m(r)$". p10, Figure 13. The figure is similar to Fig.12, but includes examples with both analog and photon counting signals. One of these examples could be used instead of Figure 12 (only analog signals) and then Fig.13 could be skipped. p9,L11, (eq.10),Authors should explain in the text that this normalization is possible only in particles free ranges. p9,L12-L14, Lidar signal can be smoothed to improve signal to noise ration. Then Fernald-Sassano-Klett inversion can be applied. Please give more details what is new here. p10, Section 4, Error sources of analog signal should be discussed in more details. References from previous studies on A/D errors and limits will be useful. Low intensity signals can be measured in photon counting mode. Please comment on reasons to use analog signal for ranges that signal is usually measured by photon counting. p11, L8-L10, "Their response. . .." Please provide more details with numbers form specifications. Performance of a commercial pulse generator can be tested with fast oscilloscope first and then compared with A/D converted respond. p11, L11-L15, and p12, first paragraph. This should be a presentation of new equipment. Please provide more detailed specifications or reference if they are already published. Adding a table with specs should be useful. Please concrete which specs are unique and not available in commercial pulse generators. p11, Figure 15. The figure is not discussed in the text. p12, Figure 16 and p13, Figuure17. Please discuss if the A/D convertors are in specifications by manufacturers. How the results

can help to improve lidar signals processing? Section 6. The information and specs of the telescope, and other optical elements could be summarized in a separate paragraph. "defoci" should be "defocus" in several places in the text. p16,L16. "ZEMAX" now is "OpticStudio". Please insert the version of the software used for the simulations. p18,L4 and Figures 25,26 and 27. Please provide more information about atmospheric model used for these simulations. Is it an aerosol free standard atmosphere? p16,L19. Please explain why the obstructions of secondary mirror are neglected. Is it neglected for Fig23. only? . p21,L2. "-1200/70=-17" should be "-1200/70≈-17" p21,L7 (eq14). The term " $((\lambda\_0-\lambda)/\lambda\_0)$ " should be " $((\lambda\_0\hat{ }2-\lambda\hat{ }2)/(\lambda\_0\hat{ }2))$ " p22,Figure 29D. The legend is missing. The number of the curves could be reduced for this plot. Section 7, General. It is hard to read this section as an independent text. To follow the text one need to read together with another paper referred as Freudenthaler (2016a) in reference list. Definitions of parameters and variables should be provided in a little more details. Text in this section could include few of the key equations from Freudenthaler (2016a). All the values of linear depolarization ratios (LDRmeans, LDRcorr, LDRmol) are significantly smaller than theoretically calculated molecular depolarization ratios in section 9.2.1, Table 1. Can you please explain what that means? More information should be provided about how exactly are obtained uncertainties of all the values of LDR. Is it the standard deviation of LDR values in some range? p23,L24, Please provide the definition of the linear depolarization ratio (LDR). p23,L25-26. A short description of Δ90 calibration should be provided. p24,L2. ". . . so-called Rayleigh signals. . ." One can see clouds desk and aerosols in ITRayleigh and IRRayleigh signal on top left plot in Fig.31. Please provide more information about these signals. Other term for these signal should be used – a reader can easy confuse with term "Rayleigh signal" used in Section 3 (p8,L13). p24,L18. "The broken lines.. " should be "The dashed lines….". Please provide in the text the numerical values of Vplus_mean and Vminus_mean together with uncertainties, as well as the calibration range. p24,L19. At least a simple description of GHK parameters should be done. p24,L33. ". . .very clean down . . ." How clean has been the atmosphere can be quantitative checked by methods

in Section3, for example Eq.12. p25, Figure 31, All the curves (except ITRayleigh) are very noisy above 11km height, and it seems cannot provide any sufficient information. Typical bars of standard errors should be included or limit the heights to 11 km could be applied for all the plots. Vplus_mean and Vminus_mean are really hard to read (to find them I zoomed the screen to 800%). p27,Eq.15. The symbol $\beta$ is an angle here, but backscatter coefficient and other sections. p31,L5-L8, (eq.37). The statement that equation 37 is wrong should be proved and explained in details about assumptions and approximations in both approaches. I think these lines can be skipped. If authors exist to keep them then should provide more details. p32-33, Table 1 should present only significant digits.

————————————————————